# Effects of aerosol dynamics and gas-particle conversion on dry deposition of inorganic reactive nitrogen in a temperate forest

Genki Katata[1], Kazuhide Matsuda[2], Atsuyuki Sorimachi[3], Mizuo Kajino[4], and Kentaro Takagi[5]

[1]Institute for Global Change Adaptation Science (ICAS), Ibaraki University, Ibaraki, 310-8512, Japan
[2]Tokyo University of Agriculture and Technology, 3-5-8 Saiwai-cho, Fuchu, Tokyo 183-8509, Japan
[3]Department of Radiation Physics and Chemistry, Fukushima Medical University, 1 Hikarigaoka, Fukushima, Fukushima 960-1295, Japan
[4]Meteorological Research Institute, Japan Meteorological Agency, Tsukuba, Ibaraki 305-0052, Japan
[5]Teshio experimental forest, Field Science Center for Northern Biosphere. Hokkaido University, Toikanbetsu, Horonobe, Hokkaido 098-2943, Japan

Correspondence: Genki Katata (genki.katata.mirai@vc.ibaraki.ac.jp)

**Abstract.** Dry deposition has an impact on nitrogen status in the forest environments. However, the mechanism for the high dry deposition rates of fine nitrate particles ($NO_3^-$) observed in forests remains unknown and is thus a potential source of error in chemical transport models (CTMs). Here, we modified and applied a multi-layer land surface model coupled with dry deposition and aerosol dynamic processes for a temperate mixed forest in Japan. This represents the first application of such a model to ammonium nitrate ($NH_4NO_3$) gas-particle conversion (gpc) and aerosol water uptake of reactive nitrogen compounds. Thermodynamics, kinetics, and dry deposition for mixed inorganic particles are modeled by a triple-moment modal method. Data for inorganic mass and size-resolved total number concentrations measured by a filter-pack and electrical low-pressure impactor in autumn were used for model inputs and subsequent numerical analysis. The model successfully reproduces turbulent fluxes observed above the canopy and vertical micrometeorological profiles noted in our previous studies. The sensitivity tests with and without gpc demonstrated clear changes in the inorganic mass and size-resolved total number concentrations within the canopy. The results also revealed that within-canopy evaporation of $NH_4NO_3$ under dry conditions significantly enhances the deposition flux of fine $NO_3^-$ and $NH_4^+$ particles, while reducing the deposition flux of nitric acid gas ($HNO_3$). As a result of the evaporation of particulate $NH_4NO_3$, the calculated daytime mass flux of fine $NO_3^-$ over the canopy was 15 times higher in the "gpc" scenario than in the "no gpc" scenario. This increase caused high contributions from particle deposition flux ($NO_3^- + NH_4^+$) to total nitrogen flux over the forest ecosystem (~39 %), although the contribution of $NH_3$ was still considerable. A dry deposition scheme coupled with aerosol dynamics may be required to improve the predictive accuracy of chemical transport models for the surface concentration of inorganic reactive nitrogen.

## 1 Introduction

The dry deposition of inorganic reactive nitrogen gas (e.g., $HNO_3$ and $NH_3$) and particles (e.g., $NO_3^-$ and $NH_4^+$) is one of the major pathways of nitrogen input into forest ecosystems. In East Asia, air pollutant emissions continue to increase (EANET, 2016) and, although dry deposition of inorganic reactive nitrogen has been suggested as an important consequence by prior

studies using indirect estimates (e.g., Pan et al., 2012; Li et al. 2013; Xu et al., 2015), direct measurement studies remain limited (Nakahara et al., 2019). Recent observational studies at forests revealed that the dry deposition flux of inorganic reactive nitrogen in the form of fine $NO_3^-$ was markedly higher than that expected from theory (Takahashi and Wakamatsu, 2004; Yamazaki et al., 2015; Honjo et al., 2016; Sakamoto et al., 2018; Nakahara et al., 2019). Provided that physical deposition processes are dominant, the deposition velocities of $SO_4^{2-}$ and $NO_3^-$ particles are expected to be similar because both species exist in the same sub-micron size range (e.g., Wolff et al., 2011). However, Sakamoto et al. (2018) observed deposition velocity of $NO_3^-$ as high as those of $HNO_3$ in temperate mixed forests, when using the relaxed eddy accumulation method (Matsuda et al., 2015). Nakahara et al. (2019) also observed a higher concentration gradient of fine $NO_3^-$ than of fine $SO_4^{2-}$ in cool-temperate forests using a thermodynamic equilibrium model to explain this difference via the evaporation of $NH_4NO_3$ particles in the $NH_4NO_3$-$NH_3$-$HNO_3$ triad within the canopy. Numerical chemical transport models (CTMs) for the East Asian region have also demonstrated that models can overestimate the total (gas + particle) nitrate concentrations present in many locations (Kajino et al., 2013; Shimadera et al., 2018; Morino et al., 2015; Sakurai et al., 2015). Despite many uncertain factors (e.g., emission inventory, grid resolution, and chemical and physical dynamics, and deposition modules), Shimadera et al. (2014) demonstrated that the surface concentration of total nitrate could be reproduced by increasing the dry deposition velocity of $HNO_3$ by a factor of 20 with respect to previous studies. Hence, the deposition velocity of $NO_3^-$ in fine particles and/or $HNO_3$ are among the major uncertainties in the CTMs.

Modeling studies for dry deposition have demonstrated the importance of the volatilization of $NH_4NO_3$ during the dry deposition of gaseous and particulate nitrates (Brost et al., 1988; van Oss et al., 1998; Kramm and Dlugi, 1994), although the process itself has been known in the atmospheric chemistry community for some time (Seinfeld and Pandis, 2006). Such modeling studies used the "big-leaf" concept to represent the forest canopy under steady-state and thermodynamic equilibrium assumptions, with chemical reaction rates provided by observational data. Nemitz and Sutton (2004) developed a more sophisticated model through the introduction of chemical timescales for a size-resolved particle, and showed that the thermodynamic gas-particle conversion of $NH_4NO_3$ (hereafter referred to as gpc) may explain the bi-directional fluxes observed above a Dutch heathland. However, several uncertainties remained in their modeled results, largely due to uncertainties in empirical treatments of particle deposition and thermodynamic processes, and the assumption that gas concentrations are in equilibrium with the pure $NH_4NO_3$ particle phase (Nemitz and Sutton, 2004). Meanwhile, the equilibration of aerosols with surrounding liquid water is also important in determining their thermodynamic equilibrium (e.g., Fountoukis and Nenes, 2007). An accurate evaluation of the impact of $NH_4NO_3$-$NH_3$-$HNO_3$ conversion within the canopy requires a process-based model that includes the thermodynamics of mixed inorganic aerosols linked to gas-phase chemistry, while ensuring that no equilibrium assumption is required (Nemitz, 2015).

We here propose a new multi-layer land surface model coupled with dry deposition and aerosol dynamics (thermodynamics and kinetics of mixed inorganic aerosols) for forest environments. In aerosol dynamics modeling, the moment method is used to reduce computational costs and incorporates general processes, such as condensation, coagulation, below-cloud scavenging (e.g., Binkowski and Shankar, 1995), and dry deposition (Bae et al., 2009). These processes are implemented into a multi-layer atmosphere-SOiL-VEGetation model (SOLVEG) including particle and fog droplet deposition and hygroscopic particle

growth processes (Katata et al., 2014). We apply the model to a Japanese mixed forest for calibration and validation. Finally, we use numerical experiments to examine the impacts of two key processes on dry deposition flux over the canopy: gas-particle conversion of inorganic nitrogen compounds and hygroscopic growth.

## 2 Model description

### 2.1 Model overview

A one-dimensional multi-layer SOLVEG model consists of four sub-models: atmosphere, soil, vegetation, and radiation within the vegetation canopy. A general description of gas and particle transport and dry deposition is given in Katata and Ota (2017). In the atmosphere sub-model, one-dimensional diffusion equations are solved between atmospheric layers for horizontal wind speed, potential temperature, specific (relative) humidity, liquid water content of fog, turbulent kinetic energy and length scale (Katata, 2009), and gas and particle concentrations (Katata and Ota, 2017). Observational data are used to determine the upper boundary conditions. Bulk transfer equations are applied at the lowest layer using soil surface temperatures and specific humidity calculated by the soil sub-model.

In the soil sub-model, soil temperature, volumetric soil water content, and soil pore specific humidity are predicted from heat conduction, mass balance in liquid water, and water vapor diffusion equations, respectively (Katata, 2009). Root water uptake is calculated from the transpiration rate in the vegetation sub-model. Mass conservation equations for the liquid and gas phases are solved to determine soil $CO_2$ (Nagai, 2004). Organic matter dynamics are also considered (Ota et al., 2013): microbial decomposition and dissolved organic carbon (DOC) leaching in the above-ground litter layer, the below-ground input of carbon from roots (root litter), and soil organic carbon (SOC) turnover and DOC transport along water flows throughout the soil profile are determined for three SOC pools (active, slow, and passive) with different turnover times.

In the vegetation sub-model, profiles for leaf temperature, leaf surface water, and vertical liquid water flux are predicted (Nagai, 2004). The heat budget equation at the leaf surface uses key variables from the atmosphere sub-model combined with the radiation scheme to predict leaf temperature. At the upper boundary of the sub-model, a given precipitation intensity is used to calculate the vertical liquid water flux within the canopy based on the surface water budget equation. The $CO_2$ assimilation rate due to photosynthesis is predicted using Farquhar's formulations (Farquhar et al., 1980) together with stomatal resistance. In the radiation sub-model, direct and diffuse downward and upward fluxes of solar and long-wave radiation are calculated to determine the radiation energy input at each canopy layer. Fractions of sunlit and shaded leaves at each canopy layer are used to calculate stomatal resistance and the total energy budget.

A multi-layer snow module is unique in its ability to include the gravitational and capillary liquid water flows in the unsaturated snow layer based on van Genuchten's concept of water flow in the unsaturated zone (c.f., Hirashima et al., 2010; Katata et al., 2020). In the soil module, freeze-thaw processes based on the freezing-point depression equation are considered in equations of heat conduction and liquid water flow. Winter-related processes for grassland phenology, such as leaf development and senescence due to cold stresses, are also implemented in the vegetation sub-model. Carbon gain, both from photosynthesis and remobilized reserves, is allocated to sinks according to changing sink priorities and strengths. Sink strengths are calculated

based on the dynamics of leaves and stems and their acclimation to low temperatures. The removal of tillers and leaves by cutting can also be simulated during the growing season, as can subsequent regrowth of the sward. The regrowth rate after cutting is calculated at each phenological stage. The natural turnover of leaves and roots is modeled using typical life spans in years. Rooting depth and the fraction of roots in soil layers are modeled as functions of root biomass. The daily amount of dead root biomass (root litter) is used as the input to SOC in SOLVEG's soil sub-model.

## 2.2 Dry deposition

Since full descriptions for the dry deposition process of gases and particles are available in Katata et al. (2013; 2014) and Katata and Ota (2017), this subsection presents only the key equations.

Using compensation points for trace gases in the sub-stomatal cavity, $\chi_s$ (nmol m$^{-3}$), and above the leaf water surface, $\chi_d$ (nmol m$^{-3}$), we model bi-directional gas exchange fluxes with stomata, $F_{gs}$ (nmol m$^{-2}$ s$^{-1}$), and with leaf water surfaces, $F_{gd}$ (nmol m$^{-2}$ s$^{-1}$), for each canopy layer following the approach presented in Katata et al. (2013):

$$F_{gs} = a(D_{gas}/D_w)R'^{-1}[(r_b + r_d)\chi_s - r_b\chi_d - r_d\chi_a], \tag{1}$$

$$F_{gd} = a(D_{gas}/D_w)R'^{-1}[(r_b + r_s)\chi_d - r_b\chi_s - r_s\chi_a], \tag{2}$$

where $a$ is the leaf area density (m$^2$ m$^{-3}$), $D_{gas}$ and $D_w$ are the diffusivities (m$^2$ s$^{-1}$) of trace gas and water vapor, respectively, $r_b$, $r_s$, and $r_d$ are the resistances (s m$^{-1}$) for leaf boundary layer, stomata, and the evaporation (cuticular), $\chi_a$ is the ambient gas concentration (nmol m$^{-3}$) in the canopy layer, and $R' = (r_b r_s + r_b r_d + r_s r_d)$. The total gas exchange flux over the leaves can be calculated as the sum of $F_{gs}$ and $F_{gd}$ for all canopy layers. In accordance with a number of observations (e.g., Huebert and Robert, 1985), all $\chi_s$, $r_d$, and $r_s$ are set at zero for highly reactive and water-soluble gas species of HNO$_3$ and HCl, i.e., we assume perfect absorption by plant canopies. For both species, parameterization specific to a deciduous forest by Meyers et al. (1989) is used to calculate $r_b$. For NH$_3$, $\chi_s$ is calculated based on the thermodynamic equilibrium between NH$_3$ in the liquid and gas phases (Nemitz et al., 2000; Sutton et al., 1994) as follows:

$$\chi_s = \frac{161500}{T_c}\exp\left(\frac{10378}{T_c}\right)\Gamma_s, \tag{3}$$

where $T_c$ is the canopy temperature (°C) and $\Gamma_s$ is the stomatal emission potential (also known as the apoplastic ratio) at 1013 hPa (Nemitz et al., 2004). NH$_3$ concentrations in leaf surface water ($\chi_d$) are calculated by assuming Henry's Law and dissociation equilibria with atmospheric concentrations of NH$_3$ at each canopy layer. To calculate the exchange flux of SO$_2$ and NH$_3$ over the wet canopy, an empirical formula for $r_d$ is applied (Massad et al., 2010) as follows:

$$r_d = 31.5AR^{-1}\exp[b(100 - RH)], \tag{4}$$

where $b$ is the constant, $RH$ is the relative humidity (%) and $AR$ is the ratio of total acid/NH$_3$, represented as (2[SO$_2$] + [HNO$_3$] + [HCl])/[NH$_3$] in each atmospheric layer. The value of $AR$ is determined from the gaseous inorganic concentration. Since the affinity (such as solubility in water) of SO$_2$ at the leaf surface is approximately twice that of NH$_3$ (van Hove et al., 1989), a half value of $r_d$ calculated by Eq. (4) is applied to SO$_2$ deposition. The $RH$ value could be affected by leaf surface

water content at each canopy layer as a function of water balance due to leaf surface water evaporation, the interception of precipitation by leaves, capture of fog water by leaves, and drip from leaves (Katata et al., 2008; 2013). Since our model is not a dynamic modeling approach (e.g., Sutton et al., 1998; Flechard et al., 1999), uncertainties in Eq. (4) could affect the simulation of charging and discharging of $NH_3$ in the cuticle due to the wet canopy being in equilibrium with non-zero leaf surface concentrations of $NH_3$.

The gas exchange flux over the ground $F_{g0}$ (nmol m$^{-2}$ s$^{-1}$) was described with compensation points at the ground $\chi_g$ (nmol m$^{-3}$) as follows:

$$F_{g0} = (D_{gas}/D_w)c_{H0}|\mathbf{u_r}|(\chi_{a0} - \chi_g), \tag{5}$$

$$\chi_g = \begin{cases} \frac{161500}{T_{s0}}\exp\left(\frac{10378}{T_{s0}}\right)\Gamma_g & (for NH_3) \\ 0 & (\text{for other gases}) \end{cases}, \tag{6}$$

where $c_{H0}$ is the surface exchange coefficient for heat, $|\mathbf{u_r}|$ and $\chi_{a0}$ are the horizontal wind speed (m s$^{-1}$) and gas concentration at the bottom of the atmospheric layer (nmol m$^{-3}$), respectively, $T_{s0}$ is the soil surface temperature (°C), and $\Gamma_g$ is the ground level emission potential for $NH_3$ at 1013 hPa.

As explained in Katata et al. (2014), the particle deposition rate of each inorganic species in each canopy layer, $F_p$ ($\mu$g m$^{-2}$ s$^{-1}$ or # m$^{-2}$ s$^{-1}$ ), is represented as follows:

$$F_p = aE_p(D_p), \tag{7}$$

$$E_p = \varepsilon(D_p)F_f|\mathbf{u}|c_p(D_p), \tag{8}$$

where $E_p$ is the particle capture by leaves ($\mu$g m$^{-3}$ s$^{-1}$ or # m$^{-3}$ s$^{-1}$), and $\varepsilon$ the total particle capture efficiency by leaves as a result of inertial impaction (Peters and Eiden, 1992), gravitational settling, Brownian diffusion (Kirsch and Fuchs, 1968), and interception (Fuchs, 1964; Petroff et al., 2009). $F_f$ is the shielding coefficient for particles in the horizontal direction, $|\mathbf{u}|$ is the horizontal wind speed (m s$^{-1}$) at each canopy layer, and $c_p$ is the particle mass or number concentration ($\mu$g m$^{-3}$ or # m$^{-3}$). $E_p$, $\varepsilon$, and $c_p$ are integration values of given size bins with particle diameters ($D_p$ [$\mu$m]).

## 2.3 Aerosol dynamics

In order to simulate changes in particle size due to condensation, evaporation, and water uptake, a triple-moment modal method (Kajino et al., 2012) is employed at each atmospheric layer in SOLVEG. Particles are grouped into fine (accumulation) and Aitken mode with size distributions prescribed by a lognormal function; the coarse mode is not considered in the simulation. The lognormal function is identified by three parameters: number concentration ($N$ [# m$^{-3}$]), geometric mean diameter ($D_g$ [$\mu$m]), and geometric standard deviation ($\sigma_g$). The triple-moment method predicts spatiotemporal changes in three moments ($k$) in order to identify changes in the shape of the lognormal size distribution of each mode. The selected three moments are 0th, 2nd, and 3rd ($M_0$, $M_2$, and $M_3$), which are respectively number ($N$), surface area (m$^2$ m$^{-3}$), and volume concentrations (m$^3$ m$^{-3}$). $D_g$ values for each moment are named $D_{g0}$, $D_{g2}$, and $D_{g3}$. The relationships of the above lognormal parameters

with the three moments for each atmospheric layer are as follows:

$$M_k = ND_{g0}^k \exp\left[\frac{k^2}{2}\ln^2\sigma_g\right], \tag{9}$$

$$D_{g0} = \left[\frac{M_2}{M_0}\right]^{\frac{3}{2}}\left[\frac{M_3}{M_0}\right]^{-\frac{3}{2}}, \ln^2\sigma_g = -\ln\left[\frac{M_2}{M_0}\left(\frac{M_3}{M_0}\right)^{-\frac{2}{3}}\right]. \tag{10}$$

Particle growth is dynamically solved following the method of Kajino et al. (2012). Gas to particle mass transfer is driven by
the difference between the current state and the thermodynamic equilibrium state, as simulated by the ISORROPIA2 model
(Fountoukis and Nenes, 2007) for semi-volatile inorganic components such as $NO_3^-$, $NH_4^+$, $Cl^-$, and liquid water ($H_2O$).
The gas-phase chemical production of $HNO_3$ could further affect simulated $HNO_3$ concentration and flux, and this parameter
should therefore be implemented to this model in the future. In the present, the gas-particle conversion of organics is not
considered because the required observational speciation data were not available. Thus, both organics and other components
of the total mass were assumed to be hydrophobic aerosols in the present simulation. Since the current study focuses on mass
gain/loss specific to accumulation mode aerosols, coagulation processes are also not included. Brownian coagulation, while
critically important for predicting the number concentration of Aitken mode particles, is not important in the prediction of
accumulation mode particle mass (e.g., Kajino et al., 2013).

## 3 Simulation setup

### 3.1 Study site and observational data

We used measurements from an observation tower in a mixed forest, namely the Field Museum Tamakyuryo (FM Tama) of the
Tokyo University of Agriculture and Technology, located in a western suburb of Tokyo, Japan (35°38'N, 139°23'E). Deciduous
trees (*Quercus* spp.) are dominant around the meteorological tower together with Japanese cedar trees (*Crytomeria japonica*).
The canopy height around the tower is approximately 20 m. The growth period of deciduous trees is typically from April to
December. Further detailed descriptions of this site are provided by Matsuda et al. (2015) and Yamazaki et al. (2015).

Simulations were carried out over two experimental periods: the first in early autumn (26 September to 11 October 2016) and
the second in late autumn (7 November to 7 December 2016). In the early autumn period, daytime (8:00-17:00 in local time)
and nighttime (17:00-8:00 in local time) mean concentrations of inorganic gases were available at five heights (1, 8, 16, 23,
and 30 m), with fine particle mass concentrations observed using a 4-stage filter-pack sampling system. System specifications
were identical to those used in Nakahara et al. (2019), except for the particle filter material. This study used a glass fiber filter
coated with Teflon for collecting fine and coarse particles. For the early autumn period, filter-pack sampling was continuously
performed during the day and night except during periods of rain. As a result, 5 daytime reading data sets and 6 nighttime
reading data sets were collected. The gaps between data in rain days of the early autumn period were linearly interpolated for
simulations. Since this interpolation could cause unrealistic effects on the results, we used only the calculations and measure-
ments in the periods of no rain for comparisons of inorganic mass concentration. For the late autumn period, the time resolution
was relatively low as weekly continuous measurements were used. After the samples were collected, inorganic ions in each

filter were extracted into deionized water by ultrasonic extraction, then analyzed using ion chromatography (Dionex ICS-1100, Thermo Scientific).

In the late autumn period, measurements of particle number concentrations were taken during the daytime (10:00-16:00) for 7 days without rainfall. Airborne particle number concentrations were measured by an electrical low pressure impactor (ELPI+, Dekati Ltd.). This involves sampled particles being charged by corona discharge and later separated by size using the principle of inertial classification through a 13-stage cascade low-pressure (40 hPa) impactor combined with a back-up filter stage. During collection, charged particles produce a current proportional to their respective number concentrations. The broad particle size distribution measured by the ELPI+ ranges from 6 nm to 10 $\mu$m. More details on the ELPI+ system used are provided in Järvinen et al. (2014). The ELPI+ particle sample inlets were placed at heights of 30, 23, 17, 8, and 1 m at the tower through TYGON intake tubing with an inner diameter of 7.94 mm and respective lengths of 6, 5, 10, 20, and 25 m . Each sampling line for the 5 measuring heights was manually switched every two minutes. The transit times for particle samples in the tubing at each height ranged between 2 and 12 s. Results of the first minute of concentration measurement were rejected in order to avoid the mixing of air samples from different heights. Data were stored at a sampling rate of 1 $s^{-1}$. Raw data were averaged over intervals of 60 s and were later used for calculating 600 s mean vertical profiles. Particle penetration efficiencies were estimated using the indoor particles in the laboratory by changing the lengths of the sampling tubes accordingly (30, 20, 15, 10, and 5 m). Based on these results, raw concentrations were corrected prior to post-processing. Furthermore, the data were further screened out according to several few selection criteria to ensure their credibility with respect to three uncertainties: uncertainty in number concentration measurements, signal to noise ratio (Deventer et al., 2015), and variation in background current (the signal obtained from particle-free air through a HEPA-filter for each particle stage) before and after the measurements.

Half-hourly meteorological data for horizontal wind speed, air temperature, and humidity at heights of 30, 25, 20, 10, 6, and 1 m at the tower were used for model input and validation. Incoming short-wave and long-wave radiation values at 30 m were used for the model input, while incoming long-wave radiation was estimated by the parameterization method of Duarte et al. (2006). Net radiation was measured using a net radiometer (Q7, REBS) and stored as half-hourly means by a data-logger (CR10X, Campbell Scientific). A sonic anemometer (81000, Young) was used to measure 3D wind velocities and air temperatures, and an enclosed infrared $CO_2/H_2O$ gas analyzer (LI-7200, Li-Cor) was used to measure the molar fraction of $CO_2$ and $H_2O$. These data were sampled at a frequency of 10 Hz using an interface unit (LI-7550, Li-Cor). Half-hourly $CO_2$, heat, and momentum fluxes were calculated using Eddy Pro software (ver. 4.2.0, Li-Cor), where double rotation (Kaimal and Finnigan, 1994) and block averaging were applied to fluctuation data in order to calculate covariance. We then corrected the effect of air density fluctuations on the flux values (Burba et al., 2012). Corrections were made for low-frequency losses (Moncrieff et al., 2004) and high-frequency losses for low-pass filtering (Ibrom et al., 2007) and sensor separation (Horst and Lenschow, 2009). All raw flux data were checked following the quality-control program of Vickers and Mahrt (1997). Finally, we applied the quality check system proposed by Mauder and Foken (2006), and excluded data judged to be low quality (qc-flag of 2).

The total (one-sided) leaf area index (LAI) measured with a plant canopy analyzer (LAI-2200, Li-cor) was 4.3 and 3.6 $m^2$ $m^{-2}$ for October and November 2016, respectively. Vertical profiles of leaf area density (LAD) were provided in order to obtain the above values of total LAI after gamma function interpolation, with a maximum at a height of 15 m following Katata et al. (2013). LAI of the understory vegetation with 0.5 m height was given a typical value of 2.0 $m^2$ $m^{-2}$ in Japanese forest (e.g., Sakai et al., 2006) due to lack of observational data.

## 3.2 Boundary and initial conditions

The boundaries of each vegetation layer were set at heights of 0.05, 0.1, 0.2, 0.3, and 0.5 m (understory vegetation), and from 1 to 20 m (forest canopy) with an increment of 1 m. Atmospheric layers were extended from the 20 m canopy to 30 m with an increment of 1 m. Half-hourly data for precipitation, atmospheric pressure, horizontal wind speed, air temperature and humidity, and incoming long- and short-wave radiation were applied to the top atmospheric layer. Inorganic mass concentrations of gases ($SO_2$, $NH_3$, $HNO_3$, and $HCl$) and PM2.5 particles ($SO_4^{2-}$, $NO_3^-$, $NH_4^+$, $Na^+$, $Cl^-$, $Ca^{2+}$, $K^+$, and $Mg^{2+}$) measured by filter-pack were linearly interpolated at half-hourly timescales. For the Aitken mode, inorganic mass concentrations were assumed to be one-tenth of those of the fine mode, based on size-resolved number concentrations from ELPI+ observations (not shown). The boundaries of the soil layers were at depths of 0.02, 0.05, 0.1, 0.2, 0.5, 1.0, and 2.0 m. Constant values for soil temperature (15 °C) and saturated volumetric water content for typical loam soil texture (0.43 $m^3$ $m^{-3}$) were used. The model setup and parameters are given in Table S1.

The boundary conditions and input data used in the setup of the simulation are summarized in Table 1. Lognormal parameter sets of ($D_{g3}$, $\sigma_g$) for fine and Aitken modes at the upper boundary condition were respectively set at (0.089 $\mu$m, 2.1) and (0.26 $\mu$m, 2.0) based on manual fitting of ELPI+ measurements at a height of 30 m (Fig. 1a). These parameter sets were applied to the early and late autumn periods. In order to simulate the vertical profiles of total number concentration within the canopy, the volume fraction of inorganic compounds, $f_{io}$, was given by the data of total inorganic mass and total number concentration. For the late autumn period, temporal changes in weekly $f_{io}$ values in both the fine and Aitken modes was shown by data from filter-pack and ELPI+ measurements. However, for the early autumn period, ELPI+ measurements were unavailable as described above. Consequently, temporal changes in $f_{io}$ (Fig. 1b) were set based on both filter-pack data at the study site and total PM2.5 mass concentrations observed at the nearest air quality monitoring station at Hachiouji (3 km west-north-west of the study site). For both periods, $f_{io}$ for the Aitken mode were assumed to be equal to the fine mode, since no observational data were available.

Since no data were available at the study site for emission potentials of $NH_3$ at the ground surface ($\chi_g$) and stomata ($\chi_s$), we used typical values of $\Gamma_g$ = 300 (Massad et al., 2010) and $\Gamma_s$ = 2000 (Neirynck and Ceulemans, 2008).

Our simulation used with basic and less time-resolved datasets for a first application of the model to the $NH_4NO_3$ gas-particle conversion and aerosol water uptake of reactive nitrogen compounds. The uncertainties associated with input data described above, such as number concentration and particle size distribution, should be improved in the future.

### 3.3 Simulation scenarios

To reveal the impacts of $NH_4NO_3$ gas-particle conversion and hygroscopic growth, the following four simulation scenarios were adopted: 1) $NH_4NO_3$ gas-particle conversion and aerosol water uptake ("gpc" scenario); 2) aerosol water uptake but no $NH_4NO_3$ gas-particle conversion ("no gpc" scenario); 3) $NH_4NO_3$ gas-particle conversion but no aerosol water uptake ("gpc dry" scenario); and 4) no $NH_4NO_3$ gas-particle conversion and no aerosol water uptake ("no gpc dry" scenario). Calculations in all scenarios were compared with observations of vertical profiles of the total number and inorganic mass concentrations within and above the canopy.

## 4 Results

### 4.1 Micrometeorology during autumn 2016

Temporal changes in friction velocity, net radiation, and sensible and latent heat and $CO_2$ fluxes over the canopy for the early autumn and the late autumn periods are shown in Figs. 2 and 3, respectively. Overall, the modeled momentum and heat fluxes agreed with observed values, although observed high friction velocities in November and December 2016 were slightly underestimated (Fig. 3a). Water vapor and $CO_2$ exchange processes determining the level of stomatal uptake of gases were also reproduced well by the model (Figs. 2 and 3; c and e).

Figures 4 and 5 illustrate time series of horizontal wind speed, air temperature, and relative humidity under the canopy in both simulation periods. Wind speed was underestimated within the forest, as was friction velocity (Fig. 3a and d), probably due to horizontal advection over hilly terrain (Matsuda et al., 2015). For calculated air temperature and humidity, the primary determinants of ambient conditions of gas-particle conversion and aerosol hygroscopic growth, calculated temporal changes were similar to observations within the canopy (Figs. 3 and 4; b and c). These features were also noted in mean vertical profiles during both daytime and nighttime in the early autumn period (Fig. S1).

### 4.2 Inorganic mass concentration and flux in early autumn 2016

Figure 6a and b shows time series for observed and calculated major inorganic nitrogen compounds ($HNO_3$ and $NH_3$ gases and $NO_3^-$, and $NH_4^+$ fine particles) under the canopy in the early autumn period. Substantial differences were observed between the "gpc" and "no gpc" scenarios for $HNO_3$ and fine $NO_3^-$ concentrations during the daytime on 28 September. In the "gpc" scenario, $HNO_3$ concentrations were increased due to the evaporation of $NH_4NO_3$ during the daytime, whereas $NO_3^-$ concentrations were decreased. Consequently, strong variations in $NO_3^-$ mass concentration were reproduced in the "gpc" scenario. A lesser impact of evaporation due to $NH_4NO_3$ on both $NH_3$ and fine $NH_4^+$ concentrations was observed.

The calculated fine mode mass-based wet diameter ($D_{g3}$) and $RH$ are shown in Fig. 6c. Hygroscopic growth has a significant impact, reflecting in particle size distribution differences between the "no gpc" and "no gpc dry" scenarios; e.g., $D_{g3}$ values in the latter were 1.4 $\mu$m higher than those in the former at 0.4 $\mu$m during the nighttime on 29 September 2016. Although this process also influences size distribution during the daytime, a competing shrinkage mechanism, $NH_4NO_3$ evaporation,

appeared in the "gpc" scenario. As a result, the difference in daytime $D_{g3}$ between the "gpc" and "no gpc" scenarios was up to in 0.12 $\mu$m on 28 September.

Figure 7 depicts vertical profiles of normalized gaseous and particulate mass concentrations in the early autumn period. The profiles averaged for the sampling periods were compared with observed profiles. In the "no gpc" scenario (Fig. 7b), vertical gradients in fine particle compounds ($SO_4^{2-}$, $NO_3^-$, and $NH_4^+$) were similar, since the same equation for collection efficiency (Eq. (8)) was used for all inorganic particle compounds. In contrast, vertical gradients of $NO_3^-$ and $NH_4^+$ concentrations drastically increased due to $NH_4NO_3$ evaporation in the "gpc" scenario (Fig. 7c and f), producing gradients similar to those observed (Fig. 7a and d). This feature was also visible in vertical profiles of mass flux for all inorganic nitrogen components during the daytime (Fig. S2). The impact of $NH_4NO_3$ evaporation was smaller during nighttime (Fig. 7c) than daytime (Fig. 7f), which also aligns with observed diurnal patterns (Fig. 7a and d). High values of observed fine $SO_4^{2-}$ concentration were calculated in both scenarios (Fig. 7a-c).

Figure 8 shows the time series for calculated apparent mass flux of $HNO_3$, $NH_3$, and fine $NO_3^-$ and $NH_4^+$ over the canopy in both the "gpc" and "no gpc" scenarios. The actual deposition flux of each component (ecosystem flux) is shown for comparison with apparent flux. The impact of $NH_4NO_3$ evaporation on fluxes was the highest from 26 to 29 September. The calculated $NO_3^-$ flux above the canopy was positive during the nighttime for several days (Fig. 8c) due to the condensation of $HNO_3$ (Fig. 8a). As for in-canopy $NH_3$ concentrations (Figs. 6 and 7), $NH_4NO_3$ evaporation has less impact on $NH_3$ flux than on other species (Fig. 8b).

## 4.3 Particle size distribution in late autumn 2016

Figure 9 shows the time series for number concentration within the canopy in the late autumn period, together with parameters for the lognormal size distribution of fine mode. Initial number concentration values on 7 November (Fig. 9a) were tuned via adjustments of the ratio of inorganic compounds ($f_{io}$) for each mode. Below-canopy $D_{g3}$ and $\sigma_g$ were smaller in "gpc dry" than in "no gpc dry" due to $NH_4NO_3$ evaporation (Fig. 9b and d), whereas below-canopy $D_{g0}$ was larger in "gpc dry" than in "no gpc dry" (Fig. 9c). For "gpc" and "no gpc" scenarios in which aerosol water was considered, $D_{g3}$ increased due to hygroscopic growth, although the influence of $NH_4NO_3$ gas-particle conversion on $D_{g3}$ was still apparent (Fig. 9b). Some discrepancies between observations and calculations were found after 25 November 2016 with respect to temporal changes in number concentration (Fig. 9a).

Figure 10 shows vertical profiles of parameters for the lognormal size distribution and normalized $NO_3^-$ mass concentration in the "no gpc" and "gpc" scenarios. For fine particles, the values of $D_{g3}$ and normalized $NO_3^-$ concentrations at 8 m were respectively 5.1 % and 8.9 % smaller in the "gpc" scenario than in the "no gpc" scenario due to evaporation of $NH_4NO_3$ (Fig. 10b and d), and the calculated $\sigma_g$ was also 1.2 % smaller (Fig. 10c). In contrast, calculated $D_{g0}$ slightly increased by 0.3 % at the same height (Fig. 10a). Almost no effect of $NH_4NO_3$ gas-particle conversion was found in the Aitken mode (Fig. 10e-h).

Figure 11 shows differences in the total number concentration for the regions above, within, and below the canopy during the daytime in the late autumn period, with particular reference to the differences among these concentrations. In the sub-micron size range (0.1-0.4 $\mu$m), differences between height pairs were strongest between 8 and 1 m (below), between 30 and 24 m

(above), and between 24 and 8 m (within the canopy). In the "no gpc dry" scenario, the difference in number concentration between height pairs was minimal in the sub-micron size range, as determined by modeled size-resolved dry deposition velocity

(Fig. 11e). Similar results have been demonstrated by past numerical studies of size-resolved particle number flux (Ryder, 2010); for particle diameter around 0.15 $\mu$m, the apparent flux switches from deposition to emission within the canopy and approximately reflects the peak in number size distribution. Furthermore, apparent emission fluxes were represented as more particles shrink into a given size bin from the next larger size bin than leave to the next smaller size bin, whereas more particles shrink out of a given size bin than shrink into it from the next larger size bin, resulting in apparent fast deposition

(Ryder, 2010). In the "gpc" scenario (Fig. 11a-d), particles in fine mode shrunk in this size range due to in-canopy $NH_4NO_3$ evaporation, resulting in an apparent tendency to emit from the canopy to the air above (Fig. 11d). Meanwhile, differences in number concentrations between 24 and 8 m (within the canopy) for large particles (> 0.3 $\mu$m) were excessively high in the "gpc" scenario compared to observational data. In the "no gpc" scenario, in which only aerosol water uptake was considered (Fig. 11c), fine particle sizes increased due to hygroscopic growth (Fig. 10), and the concentration differences between height

pairs always remained positive in this range. Finally, in the "gpc" scenario where both $NH_4NO_3$ evaporation and hygroscopic growth processes are considered (Fig. 11b), calculated negative gradients of number concentration appeared between 24 and 8 m (within the canopy) for the sub-micron range 0.1-0.4 $\mu$m, again similar to observed patterns (Fig. 11a).

## 5   Discussion

### 5.1   Uncertainties in observation and model results

SOLVEG reproduced the general features of gas concentration, fine particle mass, and fine particle number concentration observed within the canopy. Several uncertainties (e.g., low time resolution of weekly filter-pack data in the late autumn period; initialization of measurement uncertainty; complex topography of the study site) may cause underestimations in calculated wind speed (Figs. 3a and 4a) and overestimations in total number concentration within the canopy after 25 November 2016 (Fig. 9a). In Fig. 8, the conditions for $NH_4NO_3$ condensation were calculated for the studied forest, although these conditions

are normally found over strong sources of $NH_3$ (e.g. Nemitz et al., 2009). Thus, the results of this study should be considered a first test of the model to the $NH_4NO_3$ gas-particle conversion and aerosol water uptake of reactive nitrogen compounds, rather than a conclusive assessment of its capability.

Another uncertainty in the results could be associated with the assumption of the same composition in size at the initial and boundary conditions. Variations of chemical composition in size cause variations in equilibrium vapor pressure at particle

surface due to Kelvin and Raoult effects, due to uncertainty in the simulation of swelling and shrinking of particles. Since we used a modal aerosol dynamics method, the differences of these effects within each mode cannot be resolved. It is necessary to revisit this issue in the future using size-resolved composition measurements and size-resolved aerosol models, as done by Ryder (2010), to assess this uncertainty.

Concerning potential shortcomings in the modeling aspect, particle growth due to biogenic secondary organics was not

considered and might increase uncertainty in model results. Although this effect might not be important for the dry deposition

and evaporation processes that formed the main focus of this study, this effect certainly influences particle mass flux in the forest itself. Nevertheless, the order of the magnitude of observed normalized inorganic mass concentration within the canopy during the daytime, i.e., $SO_4^{2-} > NH_3 > NH_4^+ > NO_3^- > HNO_3$ (Fig. 7a) was well reproduced by the "gpc" scenario of the model (Fig. 7c). For the late autumn period, while there is no direct measurement of aerosol water content, the ambient $RH$ profile to determine hygroscopic aerosol growth was reproduced (Figs. 4c and 5c). As a result, the observed in-canopy negative gradient in number concentration (i.e., apparent emission of particles) in 0.1-0.4 $\mu$m size range was simulated in the "gpc" scenario (Fig. 11a and b). These results indicate that the model developed can be effectively used to address the impact of aerosol dynamics on dry deposition processes.

## 5.2 Formation mechanisms of particle size distributions

The complex form of the particle size distributions can mainly be explained by a combination of (1) effects of in-canopy $NH_4NO_3$ evaporation of small particles and (2) fine mode hygroscopic growth of large particles. Observed vertical gradients of size-resolved number concentrations within the canopy were reproduced only in the "gpc" scenario (Fig. 11b). Other scenarios showed different tendencies as follows (Fig. 11c-e). When only dry deposition processes were considered (Fig. 11e), number concentrations above the canopy were always larger than those within the canopy. Although the sharp negative gradient of number concentration between height pairs was computed for the 0.1-0.4 $\mu$m size range, the addition of gas-particle conversion processes to the model (Fig. 11d) caused positive gradients for large particles (> 0.2 $\mu$m) to take on excessively high values compared to observational data. The number concentration of such large particles increased within the canopy due to hygroscopic growth (Fig. 11d and e), resulting in a negative gradient from the air above the canopy to the air within the canopy.

## 5.3 Impacts of gas-particle conversion and aerosol dynamics on dry deposition

To quantify the impact of gas-particle conversion of $NH_4NO_3$ on fine $NO_3^-$ flux above the canopy, we plotted the respective ratios of $HNO_3$, $NO_3^-$ and $NH_4^+$ fluxes over the canopy in the "gpc" scenario ($F_{gpc}$) to those in the "no gpc" scenario ($F_{nogpc}$), plotting each such ratio against $RH$ at the top of the canopy in the early autumn period (Fig. 12). In this study, since the water uptake of aerosols, typically represented as the hygroscopic growth factor defined as the ratio between humidified and dry particle diameters, is almost negligible under $RH <$ approximately 80 % and increases at values over $RH > 80$ % (e.g., Fig. 6 in Katata et al., 2014), we defined a threshold of 80 % for high and low $RH$ conditions. As shown in Fig. 12b, the gas-particle conversion of $HNO_3$ and $NO_3^-$ shifted toward the particle phase under high $RH$ conditions. Conversely, under low $RH$ conditions (< 80 %), most $F_{gpc}/F_{nogpc}$ ratios were higher than unity for fine $NO_3^-$ concentrations (Fig. 12b and e). The impact of $NH_4NO_3$ evaporation on the fine $NO_3^-$ flux was very strong, i.e., just below the deliquescence relative humidity ($DRH$) of pure $NH_4NO_3$ (61.8 %). The value of the $F_{gpc}/F_{nogpc}$ ratio also reached ~40 around $RH = 50$ % (Fig. 12b). Notably, the thermodynamic equilibrium model in SOLVEG calculates the mutual $DRH$, which should not be pure $NH_4NO_3$ particles. Such high values of (apparent) $NO_3^-$ flux have also been observed in various forest types in Europe (Nemitz, 2015). These cases may be also affected by $NH_4NO_3$ evaporation near the surface.

Calculated HNO$_3$ fluxes decreased with decreasing $RH$, due to the evaporation of NH$_4$NO$_3$ (Fig. 12a). Most values of the $F_{gpc}/F_{nogpc}$ ratio for HNO$_3$ under dry conditions ($RH < 80$ %) were below 0.5, and even tended to be negative, reflecting emission from the forest to the atmosphere. Prior studies have found the same flux difference; the deposition velocity of HNO$_3$ varies from 4 to 7 cm s$^{-1}$ (Huebert and Robert, 1985; Meyers et al., 1989; Sievering et al., 2001). These velocities are often lower than theoretical maximum values or even negative, i.e., emission from the canopy (Pryor et al. 2002; Nemitz et al., 2004). High HNO$_3$ concentrations were observed within the canopy, indicated by the appearance of upward HNO$_3$ fluxes over the canopy (Pryor et al., 2002). This suggests the possibility of flux divergence due to NH$_4$NO$_3$ evaporation in the HNO$_3$-NH$_3$-NH$_4$NO$_3$ triad within the forest. This explanation has previously been suggested for other localities (Harrison et al., 1989; Sutton et al., 1993; Kramm and Dlugi, 1994; Müller et al., 1993).

NH$_4^+$ flux over the forest was less influenced by gas-to-particle conversion than NO$_3^-$ (Fig. 7c, f; Fig. 8d) because the dry deposition rates of NH$_3$ were substantially lower than those of HNO$_3$, such that the differences in deposition rates between NH$_3$ and NH$_4^+$ were much smaller than that between HNO$_3$ and NO$_3$. Indeed, the observed deposition trends for NH$_3$ and NH$_4^+$ were considerably weaker than those of HNO$_3$ and NO$_3^-$. Furthermore, although the major counter-ion of NO$_3^-$ was NH$_4^+$, that of NH$_4^+$ was not NO$_3^-$ but rather SO$_4^{2-}$. Even though the same count of molecules of NH$_3$ and HNO$_3$ was evaporated, the gross deposition rate of NH$_4^+$ appears to have been influenced mainly by (NH$_4$)$_2$SO$_4$ and/or NH$_4$HSO$_4$ as previously suggested by Nemitz (2015). The effect of NH$_4$NO$_3$ gas-particle conversion on NH$_3$ flux was even lower than on fine NH$_4^+$ (Fig. 6a) because the mass concentration of NH$_3$ was much higher.

## 5.4 Influencing the chemical transport modeling

Considering the gpc process, particle deposition could represent a very important nitrogen flux over the forest ecosystem. Comparing calculated daytime mass fluxes at 30 m height between the "no gpc" and "gpc" scenarios in the early autumn period (Fig. S2), the deposition fluxes of fine NO$_3^-$ and NH$_4^+$ were 15 and 4 times higher in the "gpc" scenario, respectively. Since there was almost no change in SO$_4^{2-}$ flux between the two scenarios, this change is found to result only from gpc. For gas species, both HNO$_3$ and NH$_3$ slightly decreased to 0.6 and 0.8 times due to the evaporation of NH$_4$NO$_3$ particles. This change in flux could be applied to that in the deposition velocity of each species. Furthermore, although particle deposition flux contributes only 5 % of the total nitrogen flux above the canopy in the "no gpc" scenario, this impact was increased to ~39 % (NO$_3^-$: 27.5 %, NH$_4^+$: 11.4 %) in the "gpc" scenario. It should be noted that contributions of NH$_3$ were still as large as 37 % of the total nitrogen flux, even in the "gpc" scenario. These results indicate that the increase of (apparent) particle deposition due to NH$_4$NO$_3$ evaporation may be important in chemical transport modeling.

Theoretical values of deposition velocity for sub-micron particles, typically ranging from 0.1-1 cm s$^{-1}$, may have no substantial impact on surface concentrations in CTMs. However, as discussed above, a high deposition velocity of fine NO$_3^-$ due to evaporation in the forest (up to 40 times the above values) may effectively remove nitrate particles from the atmosphere over the forest and leeward. If aerosol dynamics and gas-particle conversion processes can be incorporated into the dry deposition scheme of CTMs, we may improve upon or even eliminate prior studies' overestimates of the surface concentration of fine NO$_3^-$ (Kajino et al., 2013; Shimadera et al., 2014, 2018; Morino et al., 2015; Sakurai et al., 2015). Hicks et al. (2016) found

that, when modeling particle deposition velocities, the greatest uncertainty manifested in the range 0.1-1.0 $\mu$m. The cause of this uncertainty has not yet been convincingly established, although differing treatments of key particle deposition processes (e.g., turbulent diffusion) have been suggested by prior studies (Petroff and Zhang, 2010; Zhang and Shao, 2014). As demonstrated in Fig. 12b and c, the evaporation of $NH_4NO_3$ under less humid conditions may play an important role in the dry deposition of sub-micron particles.

## 6 Conclusions

A new multi-layer land surface model fully coupled with dry deposition and aerosol dynamics was developed to evaluate the impact of $NH_4NO_3$-$NH_3$-$HNO_3$ conversion in temperate forests. The model was applied to field studies of mass and number concentration profiles in a Japanese mixed forest during autumn 2016. Four model scenarios with and without $NH_4NO_3$ gas-particle conversion and/or aerosol water uptake were tested to quantify the impact of these parameters on processes of dry deposition. Overall, the model successfully reproduced micrometeorological conditions (in particular, relative humidity) within and above the canopy. When $NH_4NO_3$ gas-particle conversion processes were included in the simulation, the vertical gradients of normalized mass concentrations of nitrogen gases ($HNO_3$ and $NH_3$) and fine particles ($NO_3^-$ and $NH_4^+$) within the canopy were clearly higher than those of $SO_4^{2-}$. For particle size distribution, the observed emission tendency of total number concentration from the canopy to the atmosphere was explained by a larger effect of within-canopy evaporation of $NH_4NO_3$ than due to hygroscopic growth. As a result, the removal flux of calculated fine $NO_3^-$ from the air above the forest to the forest can increase by up to 40 times under the $DRH$ of pure $NH_4NO_3$. Similarly, the removal flux of calculated fine $NH_4^+$ can increase up to ~10 times, although calculations for fine $NH_4^+$ fluctuate strongly with $RH$. Conversely, $HNO_3$ flux over the forest can decrease by 50 % or more due to $NH_4NO_3$ evaporation, supporting the findings of previous studies. Processes of aerosol dynamics and $NH_4NO_3$-$NH_3$-$HNO_3$ conversion play a crucial role in the dry deposition of inorganic nitrogen particles in temperate forests. These processes can and should be incorporated into chemical transport models (CTMs) in order to improve the accuracy of total nitrate surface concentrations. An aerosol dynamics-dry deposition scheme simplified from that present in this study could therefore be implemented in CTMs.

*Data availability.* The output data in this study are publicly accessible via contacting the first author.

*Author contributions.* GK developed the model with support from MK, and performed the simulations using the data collected by KM, AS, and KT. GK prepared the manuscript with contributions from all co-authors.

*Competing interests.* We have no conflict of interest to declare.

We gratefully acknowledge the helpful comments and suggestions from Dr. Kentaro Hayashi at the National Institute for Agro-Environmental Sciences, Dr. Tatsuya Sakurai at Meisei University, Dr. Takeshi Izuta at Tokyo University of Agriculture and Technology, Dr. Satoru Miura at the Forestry and Forest Products Research Institute, and Drs. Makoto Tamura and Tetsuji Ito at Ibaraki University, Japan. Our thanks are also extended to Mr. Mao Xu at the Tokyo University of Agriculture and Technology, Japan, for his contribution to the filter-pack measurements. This work was partly supported by a Grant-in-Aid for Scientific Research (16H02933 and 17H01868) and Leading Initiative for Excellent Young Researchers, provided by the Japan Society for the Promotion of Science and the Ministry of Education, Culture, Sports, Science and Technology.

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

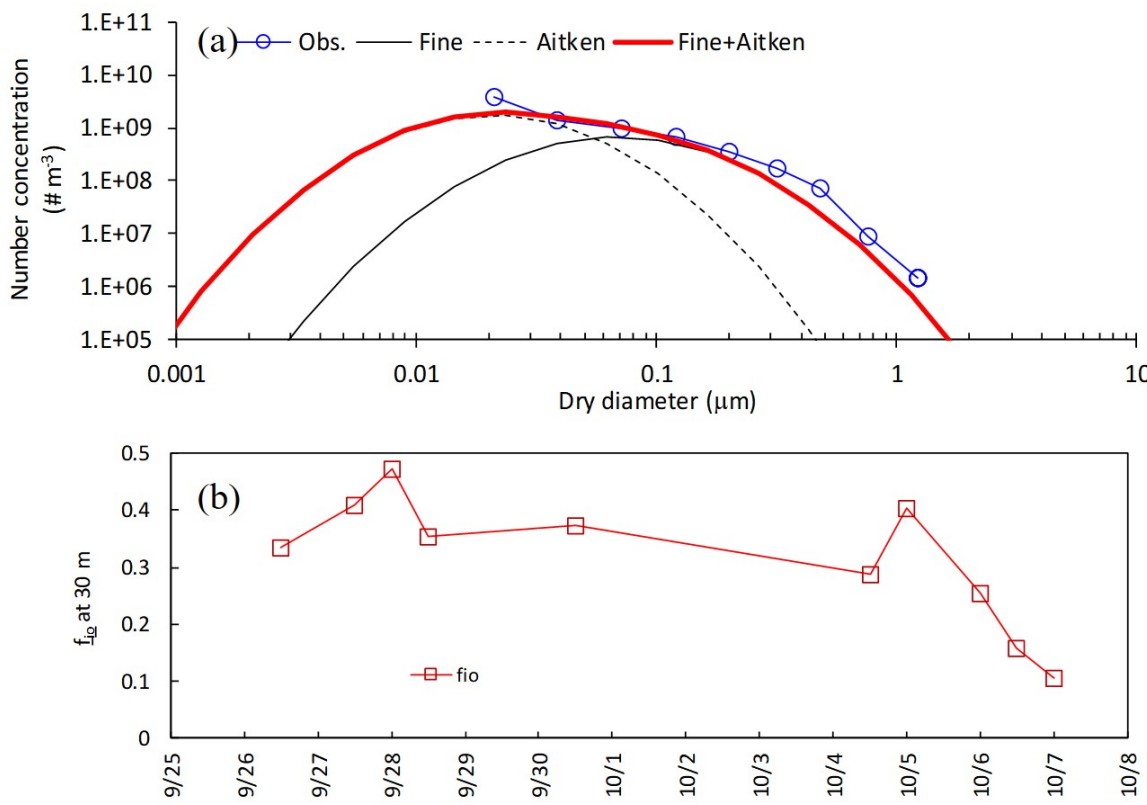

**Figure 1.** (a) Modeled and observed number-based size distribution of particles at 10:00 on 7 November 2016. (b) Temporal changes in the volume fraction of inorganic compounds ($f_{io}$) for the early autumn period.

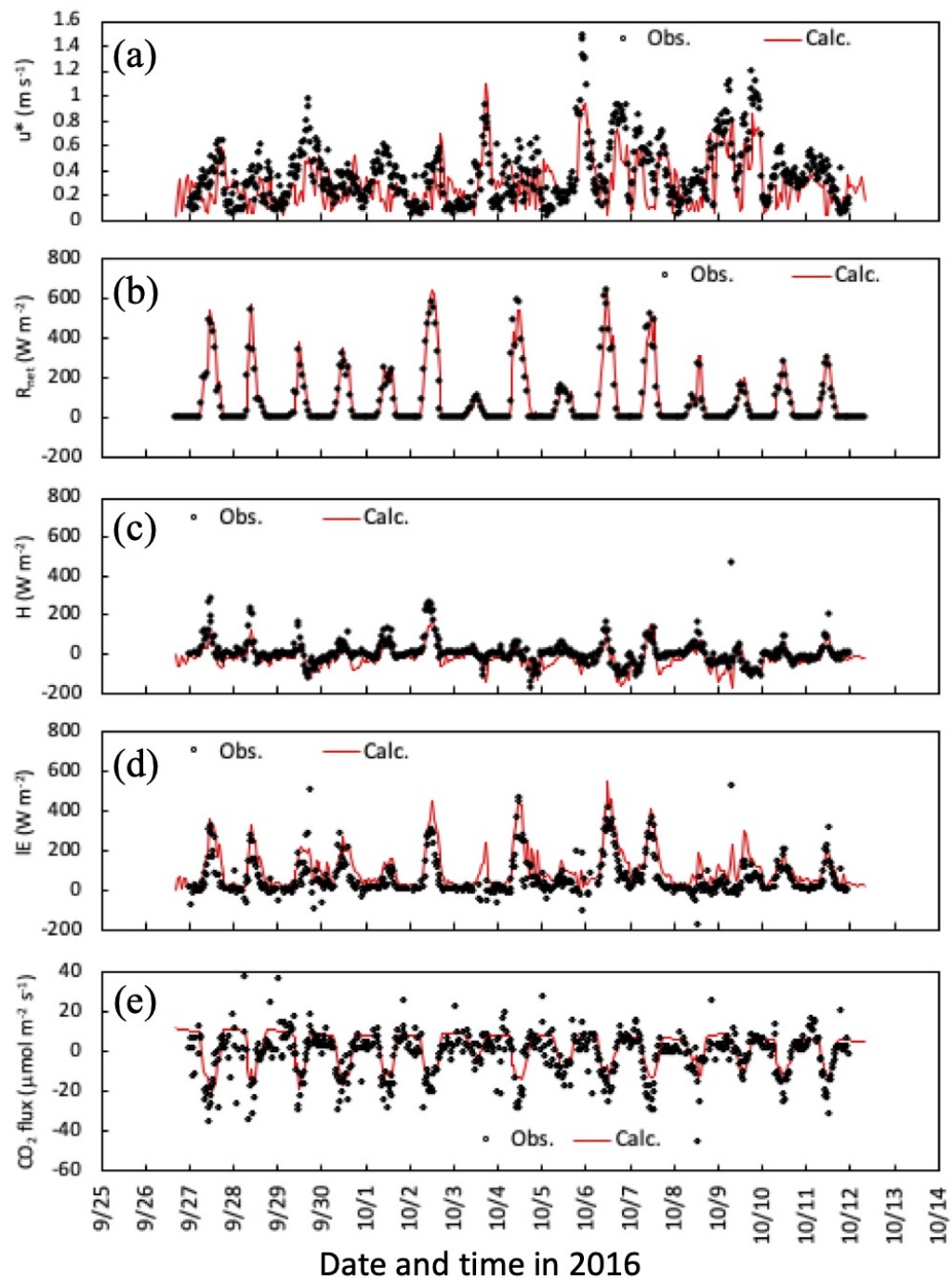

**Figure 2.** Temporal changes in observed and simulated (a) friction velocity, (b) net radiation, (c) sensible and (d) latent heat, and (e) $CO_2$ fluxes from 27 September to 11 October 2016.

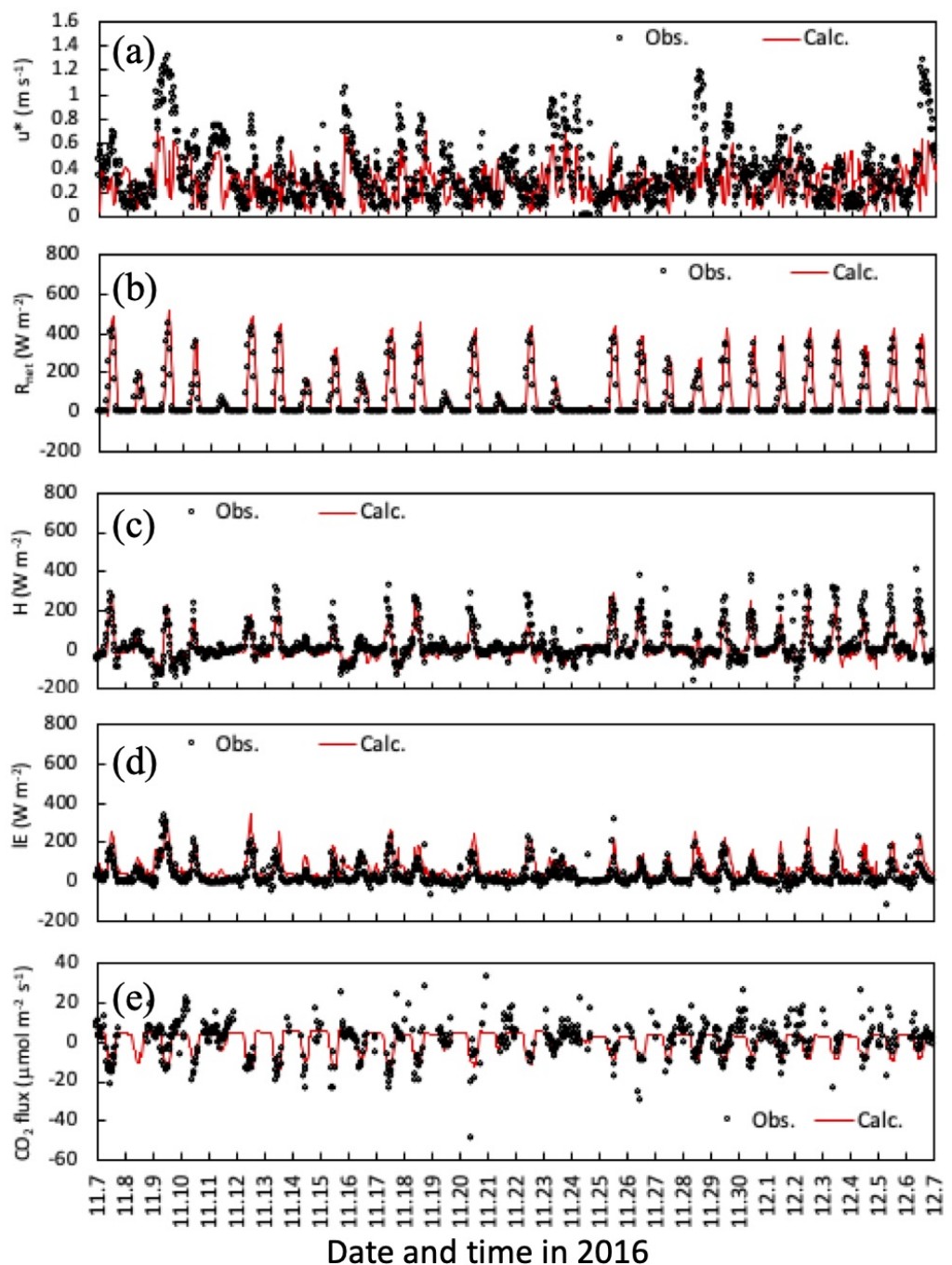

**Figure 3.** Temporal changes in observed and simulated (a) friction velocity, (b) net radiation, and (c) sensible and (d) latent heat and (e) $CO_2$ fluxes from 7 November to 6 December 2016.

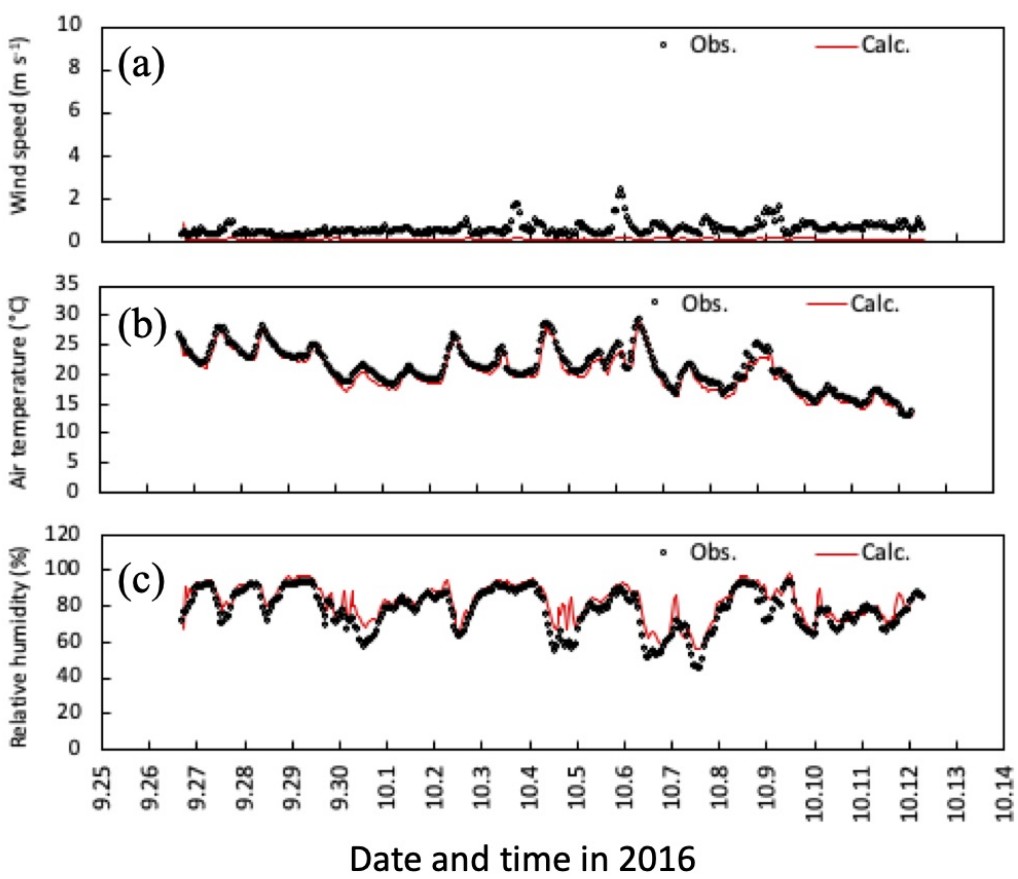

**Figure 4.** Temporal changes in observed and simulated (a) horizontal wind speed, (b) air temperature, and (c) relative humidity at 6 m height from 27 September to 11 October 2016.

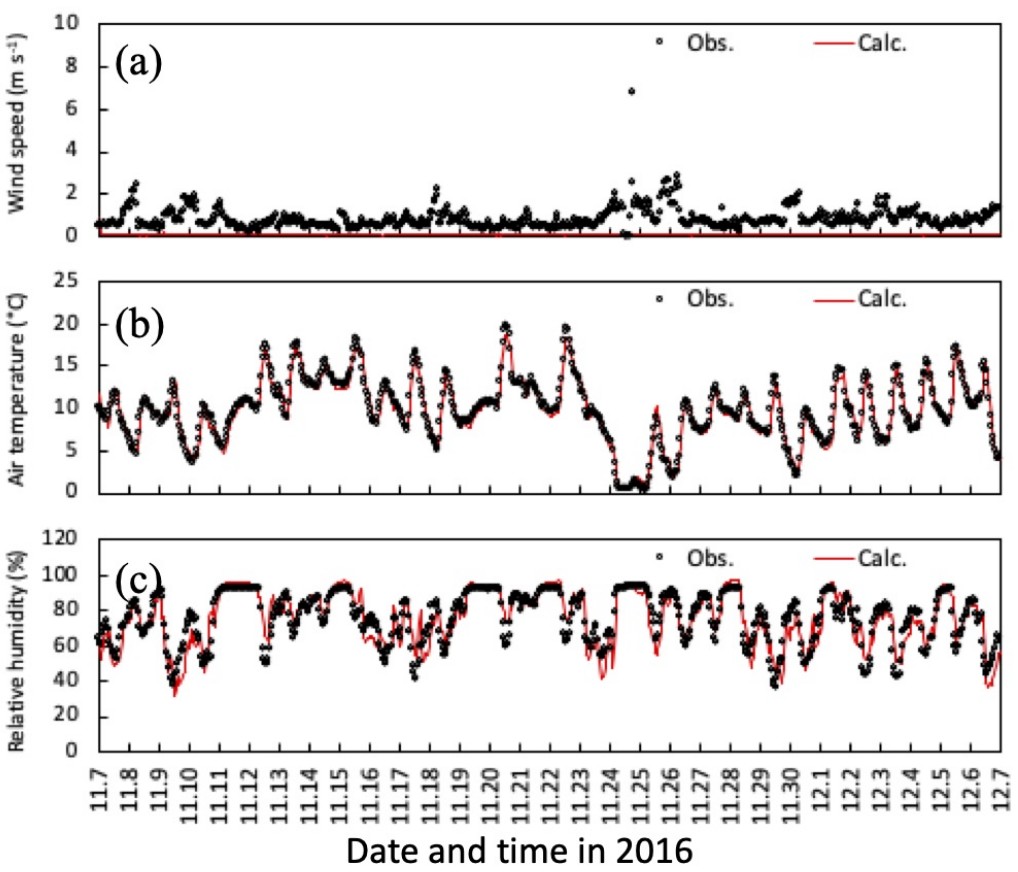

**Figure 5.** Temporal changes in observed and simulated (a) horizontal wind speed, (b) air temperature, and (c) relative humidity at 6 m height from 7 November to 6 December 2016.

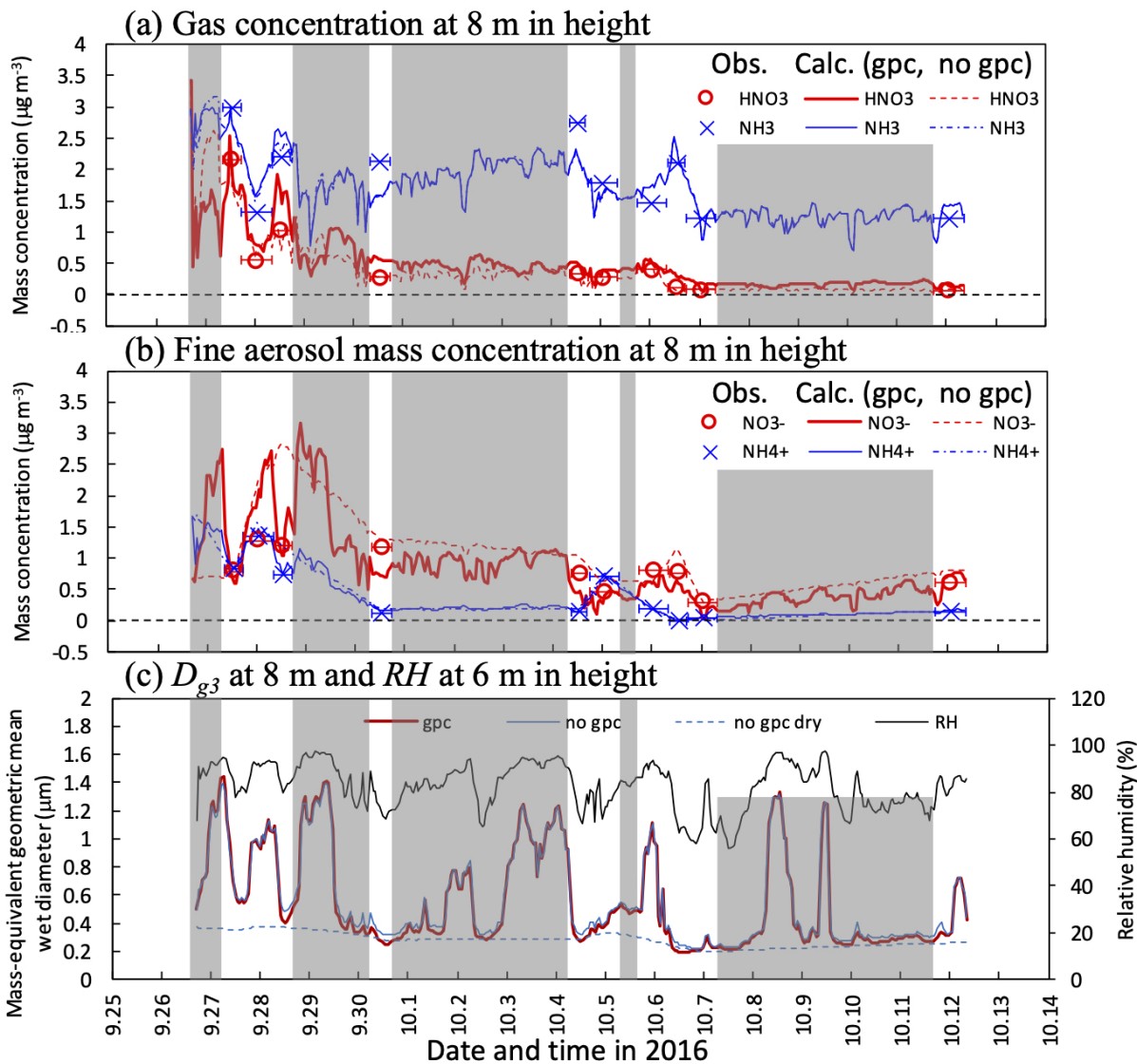

**Figure 6.** Temporal changes in observed and calculated mass concentrations of (a) $HNO_3$ and $NH_3$ gases and (b) $NO_3^-$ and $NH_4^+$ fine particles, and mass-equivalent geometric mean wet diameter ($D_{g3}$) at 8 m height from 27 September to 11 October 2016. Calculations for three scenarios ("gpc", "no gpc", and "no gpc dry", see main text for details) are plotted in the figure. The relative humidity ($RH$) calculated at 6 m height in Fig. 3c also appears in (c). Grey shaded areas represent periods of rainfalls during which no filter-pack data are available.

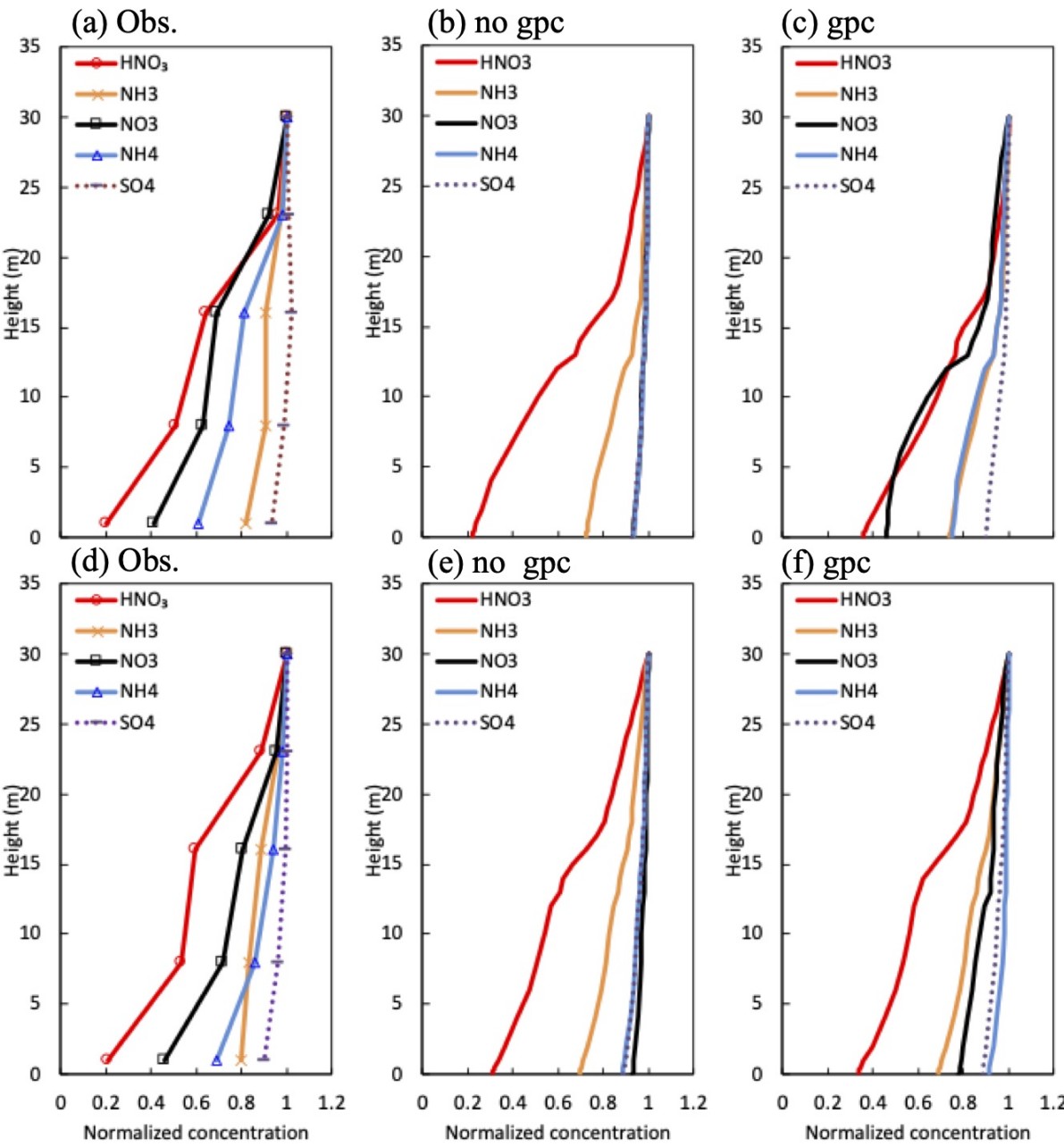

**Figure 7.** Mean vertical profiles in (a) observed and (b) calculated normalized mass concentration in the "no gpc" scenario and (c) the "gpc" scenario for $HNO_3$ and $NH_3$ gases and $SO_4^{2-}$, $NO_3^-$ and $NH_4^+$ fine particles (a-c) during the daytime and (d-f) nighttime between 27 September and 11 October 2016.

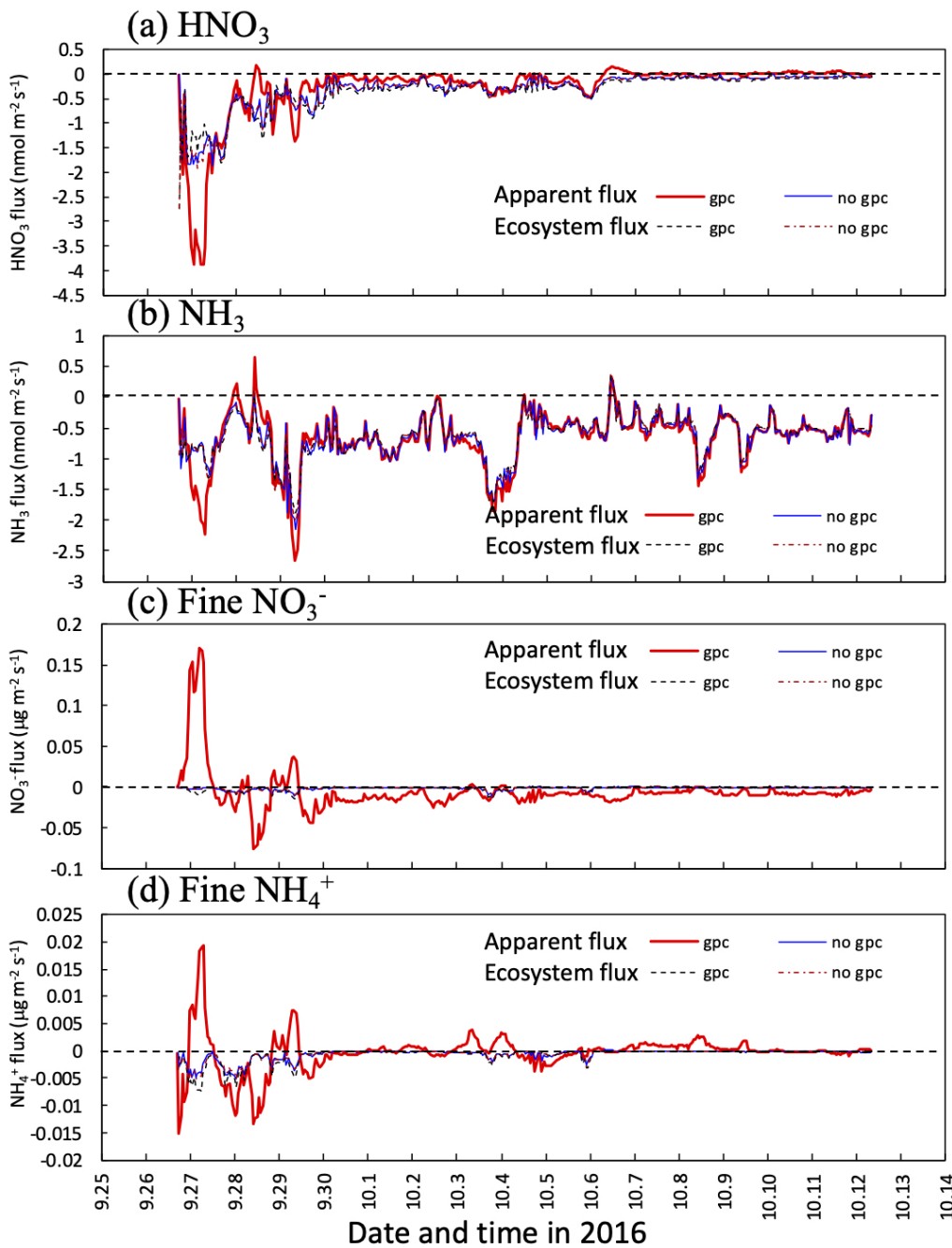

**Figure 8.** Temporal changes in calculated apparent mass flux of (a) $HNO_3$ gas and (b) $NO_3^-$ and $NH_4^+$ fine particles at 30 m height in two scenarios ("gpc" and "no gpc") between 27 September and 11 October 2016. Fluxes captured by forest (ecosystem flux) are also plotted in the figure.

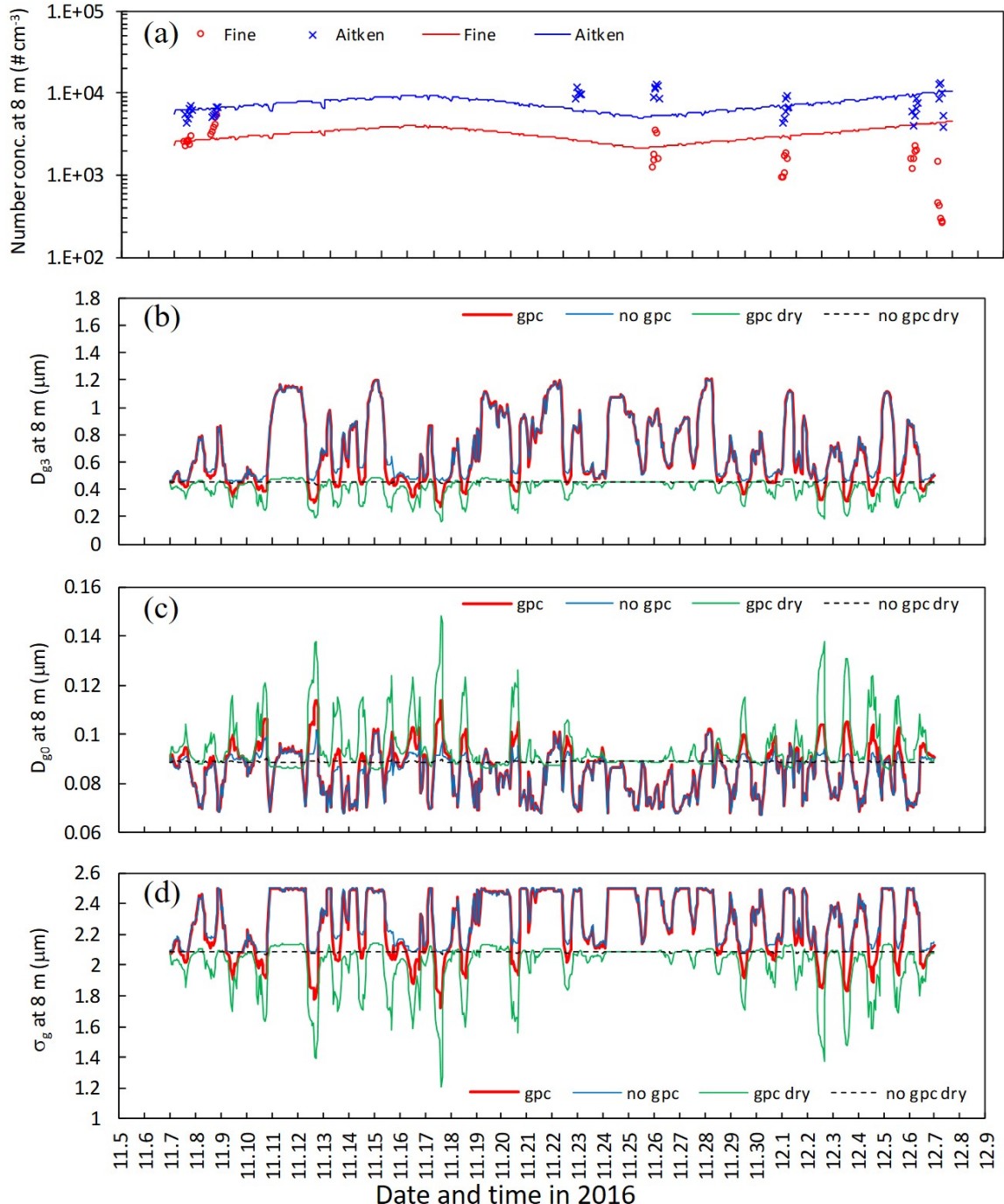

**Figure 9.** Temporal changes in (a) observed and calculated number concentration of fine and Aitken modes, and (b) calculated mass-equivalent ($D_{g3}$) and (c) number-equivalent geometric mean wet diameter ($D_{g0}$), and (d) standard deviation ($\sigma_g$) of fine particles at 8 m height from 7 November to 7 December 2016. Calculations for four scenarios ("gpc", "no gpc", "gpc dry", and "no gpc dry") are plotted in the figure.

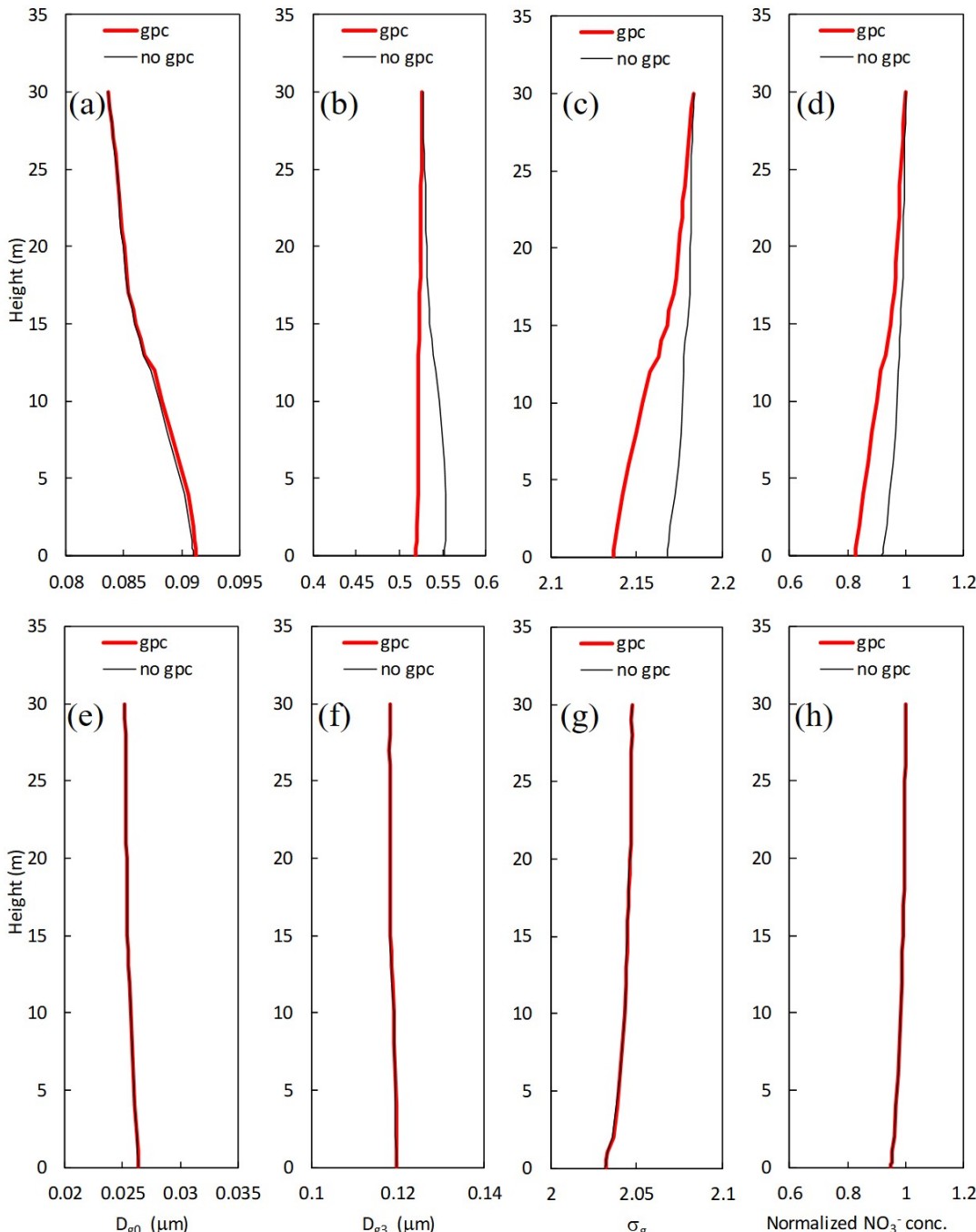

**Figure 10.** Mean vertical profiles for calculated (a, e) number-equivalent ($D_{g0}$) and (b, f) mass-equivalent geometric mean wet diameter ($D_{g3}$), (c, g) standard deviation ($\sigma_g$), and (d, h) normalized mass concentration of $NO_3^-$ for (a-d) fine and (e-h) Aitken modes in two scenarios ("gpc" and "no gpc") from 7 November to 7 December 2016.

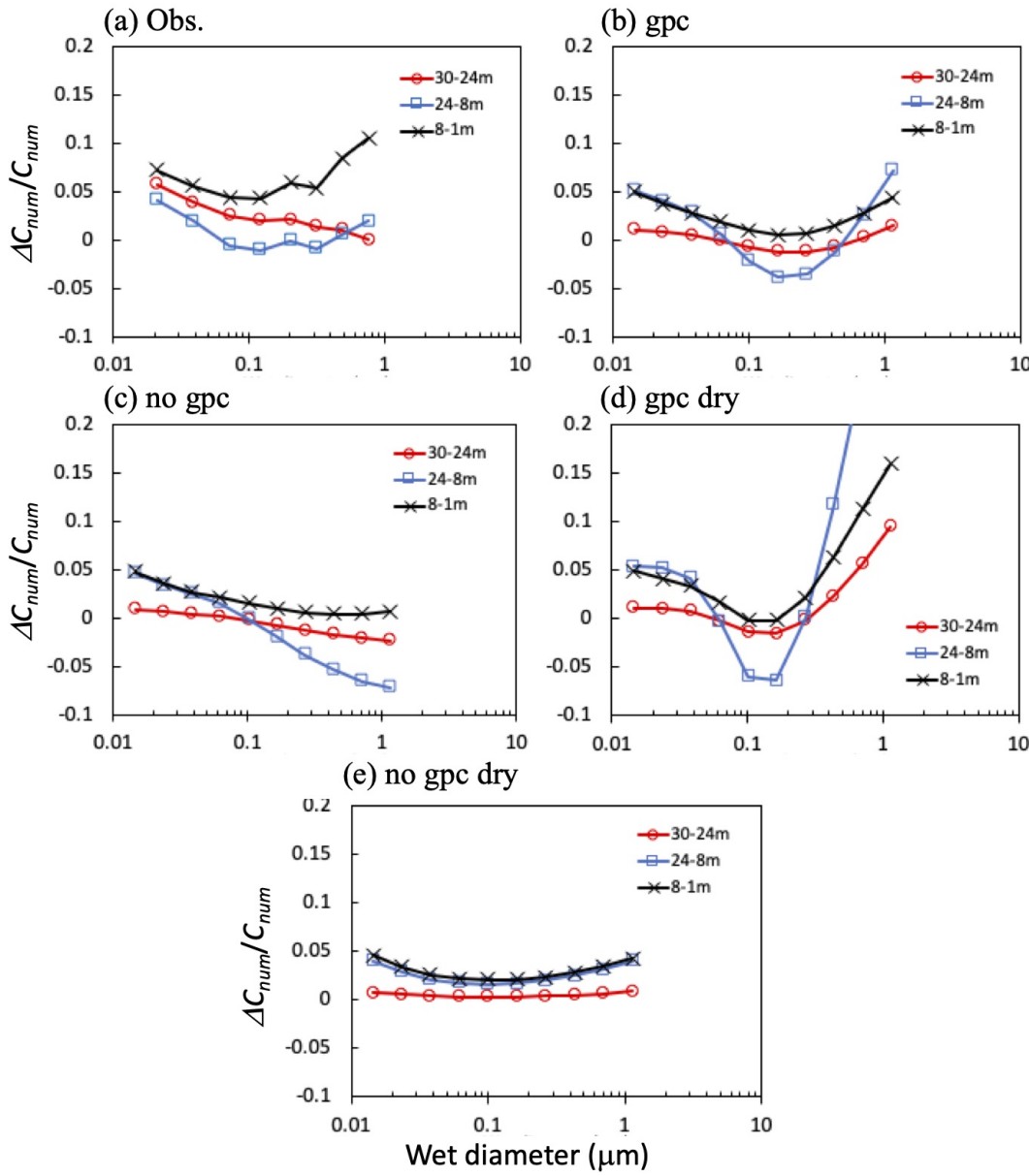

**Figure 11.** Differences ($\Delta C_{num}$) in (a) observed and (b-e) calculated mean total number concentrations ($C_{num}$) between height pairs for 11:00 -17:00 on 7, 8, 25, and 30 November 2016. Four calculation scenarios are presented in the figure: (b) "gpc", (c) "no gpc", (d) "gpc dry", and (e) "no gpc dry".

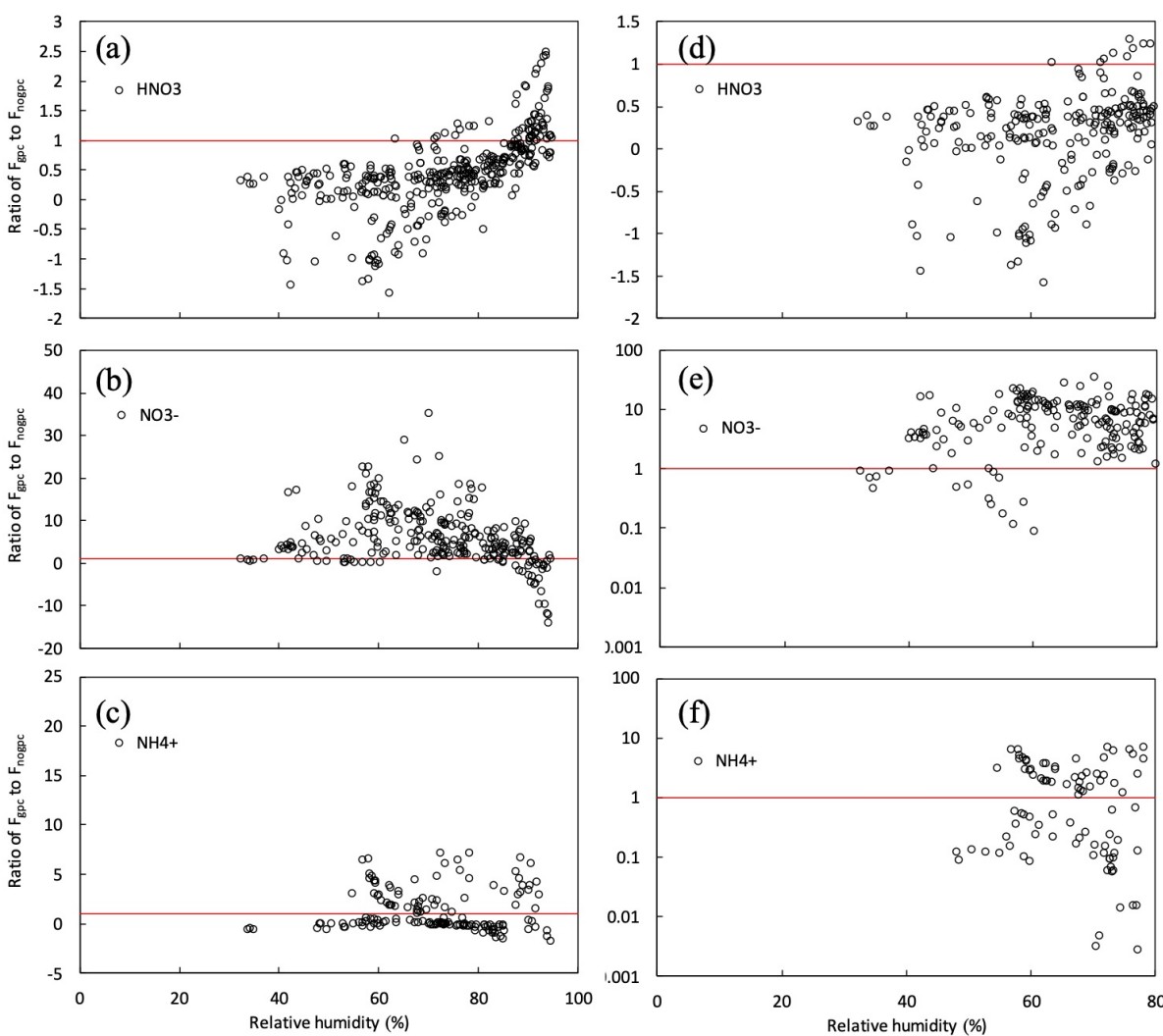

**Figure 12.** Relationship between relative humidity ($RH$) at 30 m height and "gpc" to "no gpc" ratios of calculated half-hourly fluxes ($F_{gpc}/F_{nogpc}$) for (a, d) $HNO_3$ gas and (b, e) $NO_3^-$ and (c, f) $NH_4^+$ fine particles over the canopy from 27 September to 11 October 2016. Red lines represent the situation in which $F_{gpc} = F_{nogpc}$. (d) through (f) plot the same variables as (a) through (c), but under dry conditions ($RH < 80\ \%$).

**Table 1.** Summary of input data, initial conditions, and boundary conditions for the SOLVEG simulation setup. FP: filter-pack measurements, ELPI+: ELPI+ measurements, $f_{io}$: volume fraction of inorganic compounds. It should be noted that the gaps between FP in rain days of the early autumn period were linearly interpolated for simulations.

|  | Early autumn | Late autumn |
| --- | --- | --- |
| Period | 26 September-11 October 2016 | 7 November-7 December 2016 |
| Target of simulation | Fine inorganic mass concentration/flux | Total number concentration/flux |
| Upper boundary condition |  |  |
|    Inorganic mass concentration | FC Day/night (without rain days) | FC Weekly (continuous) |
|    Particle size distribution | Fitting of ELPI+ (Fig. 1a) |  |
|    $f_{io}$ | FC and Hachiouji station | FC and ELPI+ |