# Peer review of "The effect of aerosol dynamics and gas-particle conversion on dry deposition of inorganic reactive nitrogen in a temperate forest"

_Atmospheric Chemistry and Physics, 2019_

## Referee Comment (RC1) · Anonymous Referee #1 · 9 Sep 2019

Review of

**Aerosol dynamics and gas-particle conversion in dry deposition of inorganic reactive nitrogen in a temperate forest**, by Katata et al.

**General Comments**

This work presents an analysis of size-resolved aerosol and gaseous species concentration data from a mixed forest canopy near Tokyo using a multi-layer atmosphere-soil-vegetation model with aerosol dynamics and dry deposition. In particular, the authors hope to explain observed apparent fluxes of $NO_3^-$, $NH_4^+$ and $HNO_3$ above vegetative canopies by within-canopy evaporation of ammonium nitrate ($NH_4NO_3$).

In general, I believe this to be important work and interesting, valuable data, and I also agree with their major conclusion that 3-D chemical transport models need to better incorporate the within-canopy aerosol dynamic/equilibrium processes that are the focus of this work. However, I believe that this article needs additional polishing to make its presentation more effective and the results more accessible to ACP readers. I offer my suggestions for this presentation enhancement below.

I recommend that the article be published after additional measures are taken to enhance its presentation.

**Specific Comments**

One concern is that the model used in this study is not adequately described. The paper does reference other published articles (Katata et al., 2013; Katata et al., 2014) where portions of the model are described in some detail; however, the full model used here seems to have been described in a gray literature document (Katata and Ota, 2017). It would be better if the authors included more model description in this paper, especially providing information on model setup for this particular application (e.g., model inputs, number of layers, time resolution, model outputs, etc.), referring to the other publications (or an Appendix) for details.

A second major concern is the presentation of some of the Figures, as follows:

- Figures 2 & 3 – The figures are very small and cannot be adequately evaluated, especially with respect to the agreement between the measurements and model results. Some way needs to be found to present the figures in a larger, clearer way.

- Figure 5 – The colors chosen for the vertical profiles are very difficult to distinguish between some species. Bolder color differences would be a great improvement in understanding this very important figure.

- Figure 9 – The y-axis title is so small as to be illegible. Please increase.

- Figure 10 – As mentioned below, this figure (and the discussion that goes with it) is very confusing. I don't really understand what point is being made with this figure (ratios as a

function of RH, but under "high RH conditions" and "low RH conditions" – this doesn't make sense as explained).

p. 4, lines 90-91, line 104, line 111: There seem to be two definitions of "a", one which I believe is the leaf area density, and the other a constant in Eq. (4).

p. 4, lines 90-91: The term "R'" is not defined.

p. 4, line 96: Should be "perfect absorption".

p. 4, line 104: "$T_c$" is not defined.

p. 4, Eq. (4): A reference should be provided for this formula, which is Massad et al. (2010), ACP, 10, 10359-10386.

p. 6, line 148: Was it really a "grass" fiber filter or was this supposed to be a "glass" fiber filter?

p. 6, line 150: If I understand the intention of the sentence, both instances of "reading" could (and should) be deleted – "We obtained 5 daytime data sets and 6 nighttime data sets."

p. 8, Section 3.3: The description of the simulation scenarios is somewhat confusing on first reading. The phrases "$NH_4NO_3$ equilibrium" or "no $NH_4NO_3$ equilibrium" might lead someone to believe that non-equilibrium thermodynamics is being modeled here, when actually it's just that no $NH_4NO_3$ gas-particle exchange is being allowed in the "no $NH_4NO_3$ equilibrium" scenario. A possible suggestion might be something like "$NH_4NO_3$ gas-particle conversion" and "no $NH_4NO_3$ gas-particle conversion".

p. 8, line 235: Should read "… on both $NH_3$ and fine $NH_4^+$ concentrations …"

p. 8, line 239: Should be "… competing shrinkage mechanism, …".

p. 9, line 252: Should be "… evaporation has less impact on …".

p. 9, line 269: Should be "among", not "amoung".

p. 11, first paragraph in Section 5.3: This discussion here (and Figure 10) is very confusing. How can the ratios be plotted as a function of RH, but distinctions still made between "high RH conditions" and "low RH conditions"? Whatever the subject is here, it is needs to be more clearly explained.

p. 11, line 334: There are two instances of "typically" in this sentence, which is awkward.

p. 12, in "Author contributions": Should be "… developed the model with support from MK,".

---

## Referee Comment (RC2) · Anonymous Referee #2 · 13 Nov 2019

**Review of acp-2019-703**

**"Aerosol dynamics and gas-particle conversion in dry deposition of inorganic reactive nitrogen in a temperate forest"**

**General Impression.**

The paper presents a 1-dimensional model of coupled phase-partitioning and multi-layer canopy to study the process of $NH_4NO_3$ aerosol evaporation and water uptake on surface atmosphere exchange fluxes of reactive nitrogen compounds. A number of studies have observed remarkably high apparent deposition rates of ammonium and nitrate aerosol which have been linked to aerosol evaporation processes, but few studies have attempted to model the process and thereby quantify the overall effect on N deposition. This paper presents a detailed model for this purpose and applies to reproduce concentration gradients measured above a Japanese forest. Whilst the model is detailed, the measurements are comparably basic and poorly time-resolved which limits their value in really testing the model. Nevertheless, the paper presents a useful first application of the model and makes a valuable contribution to the literature on the subject. I recommend its publication once a few concerns have been addressed as detailed below. Importantly, the description of the model and its application is very brief and important detail is missing. In addition, the authors could exploit their results a little more comprehensively.

My suspicion is that this would result in a sufficient amount of new, additional text to necessitate a re-review of the manuscript.

**Major concern**

1.  My major concern relates to the use of long-term average concentrations. Please make the time resolution of the filter pack measurements more explicit. It suggests in Line 147 that integration times were 9 hrs (day-time) and 15 hrs (night-time), whilst line 284 talks about weekly measurements. In either case I do not understand why they were used as spot measurements in the interpolation (Figs. 1 and 4). Should the model not be initialised with the same concentration for those time-intervals? Otherwise the average of the interpolated time-series does not match the measured average. Similarly, the length of the intervals of the dots in Figure 4 do not suggest that the measurements were continuous.
    Line 251 and Figure 6. I would be very surprised if the conditions for $NH_4NO_3$ condensation were met in this forest. Normally, these conditions are found over strong sources of $NH_3$ (e.g. Nemitz et al., 2009). If I understand the manuscript correctly, the authors are using long-term average concentration measurements with long-term average meteorology. In general, the vapour pressures of $NH_3$ and $HNO_3$ are dependent on temperature and humidity in a highly non-linear relationship. This means, if long-term average concentrations are paired with long-term averages in temperature and humidity, it is unlikely that the comparison of measured concentrations with the thermodynamic equilibrium concentrations evaluated with the ISORROPIA2 model can correctly assess whether there is potential for $NH_4NO_3$ evaporation of condensation. With the long-term integrated samples this problem cannot be fully resolved and as a result the entire manuscript needs to be

reformulated to some degree that full agreement between modelled and measured conditions cannot be expected. The application to this measurement dataset can only be considered a first test of the model, rather than a conclusive assessment of its capability. This problem needs to be discussed and, potentially, a sensitivity analysis could explore the uncertainty introduced by the averaging. In addition, the concentrations used for initialisation are subject to measurement uncertainty that may further limit the model / measurement comparison.

2. Overall, I am missing results and discussion on the effect of gpc (and also the equilibration with water content) on the exchange with the vegetation, in addition to the effect on the fluxes above the canopy. Only if this effect is shown to be significant, would there a need to incorporate this additional complexity into deposition schemes. To assess the importance, I would encourage the authors to quantify, from their results, the effect on the actual ecosystem flux of the various forms of N, total reactive N and also the effective bulk deposition velocity of the aerosol at the surface ($V_{ds}$), which changes because particle size changes. Presumably, this change in $V_{ds}$ is the reason for gpc changing also the in-canopy gradient of $SO_4^{2-}$ (Fig. 5) although it does not take part in the gpc process itself. The implications should be discussed.

3. Related to this, the discussion of Figure 9 is very cryptic and only accessible to those already very familiar with the subject. It is closely linked to the observations of apparent bi-directional size-segregated fluxes (e.g. Nemitz and Sutton, 2004; Ryder, 2010) and this link should be made.

**Additional scientific comments and needs for clarification:**

Abstract. It would be helpful to be more quantitative and also to include a statement on the effect on the NH3 flux as well as the total reactive N flux. By what fraction does the change in phase partitioning change the net N flux?

Line 65. Please clarify if the model also predicts the relative humidity profile which is important in controlling the phase partitioning. Related to this, Figure 3 should convey better whether the measured (in-canopy) profiles in the meteorological parameters are correctly reproduced by the model, i.e. the emphasis should be on the vertical change rather than the time-series.

Lines 100ff. I do not understand the approach taken for calculating $c_d$. The text reads as if this is chosen to match the atmospheric concentration at each canopy layer. Surely, in this case $F_{gd}$ becomes zero if the canopy layer air concentration matches the gas phase concentration in equilibrium with the $G_d$ of the leaf water layer, and $r_d$ ceases to have any effect. It is exactly the departure from equilibrium that drives the flux. Instead, $G_d$ is controlled by the previous accumulated deposition onto the leaf cuticle and the size of the water pool. How is the water pool size calculated in the model? Related to this, I am not convinced the use of an $r_d$ that is linked to acid/$NH_3$ ratio and a leaf water emission potential are internally consistent. Parameterisations of the cuticular resistance as a function of this ratio have been developed and applied within the framework of a zero leaf water emission potential (e.g. Nemitz, 2015; Fowler et al., 2009) meant to account for the effect of a non-zero leaf water concentration. By additionally introducing a non-zero $G_d$ value, this effect is accounted for twice.

Line 113. I realise that the authors are here only summarising the principles of a paper that is described in more detail elsewhere. Nevertheless, it would help to cite the approach taken to estimate the aerosol capture efficiency. Similarly, the origin of Eq. (4) needs to be mentioned.

Line 130. In addition to the ISORROPIA2 thermodynamic module, does the model treat any gas-phase chemistry? This may be important as an additional source for HNO3. If not, the authors should discuss the implications somewhere.

Line 198. It is not clear how the leaf water content was prescribed. This should affect the overall RH profile throughout the canopy and thus the results should be quite sensitive to this parameter?

Line 203. The meaning of $f_{io}$ is not clearly introduced. I understand it to be the ratio of inorganic to total aerosol mass. If so, it is constrained by the observations (in contradiction to what is stated in the manuscript) by the comparison of total inorganic aerosol mass (from the filter-pack measurements) to total aerosol mass (approximated via the ELPI+ aerosol volume). What is not constraint is its size dependence.

Section 3.2. I am missing more explanation as to how the measurements were used to drive the model and provide initial or boundary conditions. In fact the content of this section does not match its title. Did you use concentrations at a single height (if so, which one?) or several heights? How often was the model re-initialised with the measurements. Was it allowed to run to steady-state conditions or was it continuously perturbed by the measurements? Was a spin-up time used? Maybe, the input could be illustrated by adding the constraining concentrations as a top panel to Figure 4. In addition, it is numerically problematic to use the same aerosol composition across all sizes as the Kelvin effect then causes evaporation from the smallest particles and condensation on the larger ones. This would drive some of the changes in the diameters (Fig. 7), which would then not represent a response to vertical gradients but reflect inadequate initial conditions. How was this problem dealt with? In this context (and in general) I would encourage the authors to study and refer the work of Ryder (2010) who also developed a similar model and applied it to existing datasets of exchange.

Line 210. Values of $c_g$ and $c_s$ as high as 300 ppb and 2000 ppb are completely unrealistic and inconsistent with the paper of Massad et al. (2010). They also ignore the temperature dependence of Eq. (3). Do the authors mean $G_g$ = 300 and $G_s$ = 2000 (no units!)? $G_g$ is mentioned for the first time here and needs to be introduced much earlier on. Again, this calls for introducing the model in more detail than is currently done.

Section 5.1 and Line 349ff. Figure 4 suggests to me that the extrapolation of concentration into the canopy led to better agreement in the run that did NOT include gpc, whilst the text talks about an improvement. Please clarify. In general, to assess model performance, it would be much more illustrative to rearrange Figure 5 so that the three lines (Obs, no gpc, gpc) can be compared on a single plot for each compound. In addition, it would be interesting to compare a plot of how fluxes changes with height.

Line 326. I am not convinced the authors' argument here is correct. I would expect, to the first order, $HNO_3$ and $NH_3$ to be driven off the aerosol in stoichiometric ratios and thus the flux divergence for both compounds (and their aerosol counterparts) should be similar in absolute (molar) terms, independent of the deposition rate of the individual compounds. However, Fig. 5 shows normalised (i.e. relative) concentration changes and here the authors are correct with their second explanation: the relative effect on $NH_4^+$ is smaller than on $NO_3^-$ because it partly represents non-volatile sulfates.

**Technical corrections:**

Title. The word "in" does not read right in my mind. How about "The effect of aerosol dynamics and gas-particle conversion on dry deposition …"

Line 88. "fluxes with stomata … and with leaf water surfaces …"

Eqs. (1), (2), (3), (5). Please make sure the text introduces all symbols used in the equations. Many symbols (e.g. $R$, $a$, $G_s$) are not introduced.

Eq. (3) would benefit from a reference.

Eq. (5). Is the meaning of $a$ here the same as in Eqs. (1) and (2)? If not, please use a different symbol.

Eq. (5) and (6) and associated text. It would help here to make any dependence on particle diameter explicit, e.g. by writing $E_p(D_p)$.

Line 130. "… transfer is driven by the difference …"

Line 173. It is not clear to me what the word "latter" refers to. Please rephrase.

Line 190. It would be useful to state the total number of layers of the model.

Line 214. The acronym "gpc" is not introduced. Presumably it stands for gas-particle conversion? I wonder whether "thermodynamic gas-particle partitioning" would be a better concept to use throughout?

Figure 6. Please clarify in the figure caption the height for which this flux is provided as the flux is height dependent. In addition, it may be illustrative to display the actual exchange with the ecosystem under both model scenarios.

All figures: horizontal zero lines on all figures would help interpret these more easily. The font size of some labels and legends should be increased for better readability.

Line 322. This should read "2004" rather than "2004a" here, I believe.

**References:**

Nemitz, E., Dorsey, J. R., Flynn, M. J., Gallagher, M. W., Hensen, A., Erisman, J.-W., Owen, S. M., Dämmgen, U., and Sutton, M. A.: Aerosol fluxes and particle growth above managed grassland, Biogeosciences, 6, 1627–1645, https://doi.org/10.5194/bg-6-1627-2009, 2009.

Ryder, J., 2010. Emission, deposition and chemical conversion of atmospheric trace substances in and above vegetation canopies. PhD Thesis, University of Manchester, UK. Available from the University or via https://nora.nerc.ac.uk

---

## Referee Comment (RC3) · Anonymous Referee #3 · 15 Nov 2019

Review of submission ACP-2019-703 by Katata et. al.

General The author present the development of an advanced deposition model, here for the inorganic reactive nitrogen gas phase species HNO3 and NH3 and particle species NO3- and NH4+ . I suggest to use not use the term 'aerosol' when solely particles are addressed. Use the term 'aerosol' when you address particle together with the gas phase where they are dispersed in. Otherwise use 'particle'. Surely, deposition much deserves a better treatment in many atmospheric model, so in principle an improvement in deposition schemes is highly welcome. Overall, I feel the model can deliver useful results but there are many approximations it its set-up. This should be

treated most carefully. In my view, the paper needs a huge amount of improvement but I rate this as still doable and not recommend rejection. I would therefore like to recommend major revision according to all reviewer comments with external re-review necessary.

Details Abstract: I feel the abstract should give more information, at best, in a quantitative manner. It now reads too much like an introduction. What is the main numerical outcome ? What is better than before ? The abstract should clearly state what is treated. Line 13: Maybe a word should be added after 'nitrogen' ? Like 'input' ? Line 15, 16: Why is these only Japanes references, please check other deposition work. Line 29: This is not only known from / for deposition studies but also for myriad of particle characterization studies. Are there more recent references ? Line 51: What does SOLVEG mean ? Line 74: Reference Genuchten's concept Line 99: How does Eqn (3) relate to the Henry Constant ? Can you clarify more what if written in the text ? Line 104ff: Where does Eqn (4) come from ? 'Affinity' is a strange term. Better justify the approximation for SO2 deposition. Line 117 ff: See initial remark on nomenclature and revise this whole treatment consistent with clear naming. Line 139, end: . . .of the Tokyo. . .. Line 148: What is a 'grass fiber filter' ? Line 200ff: There seem to be a lot of approximations for the particle size distribution initialization. How critical can this be for the overall study ? Line 231: . . .fine particles. Lien 296: This headline must be revised. The size distribution does not have a formation mechanism, only the particles have Line 361: I think feasibility might be the wrong term. Figures: It would be great to show correlation plots for some key properties rather than only time-series plots.

---

## Author Comment (AC1) · 13 Feb 2020

Please see attached file (Supplement).

Please also note the supplement to this comment:
https://www.atmos-chem-phys-discuss.net/acp-2019-703/acp-2019-703-AC1-supplement.zip

———————————————

---

## Author Comment (AC2) · 13 Feb 2020

Please see attached file (Supplement).

Please also note the supplement to this comment:
https://www.atmos-chem-phys-discuss.net/acp-2019-703/acp-2019-703-AC2-supplement.zip

——————————————

---

## Author Comment (AC3) · 13 Feb 2020

Please see attached file (Supplement).

Please also note the supplement to this comment:
https://www.atmos-chem-phys-discuss.net/acp-2019-703/acp-2019-703-AC3-supplement.zip

---

## Author Comment (AC4) · 13 Feb 2020

Please see attached file (Supplement).

Please also note the supplement to this comment:
https://www.atmos-chem-phys-discuss.net/acp-2019-703/acp-2019-703-AC4-supplement.zip

---

## Author Response (AR1)

**Title: Aerosol dynamics and gas-particle conversion in dry deposition of inorganic reactive nitrogen in a temperate forest Authors: G. Katata et al.**

**Author response to reviewer comments**

**Response to Anonymous Referee #1**

General Comments: This work presents an analysis of size-resolved aerosol and gaseous species concentration data from a mixed forest canopy near Tokyo using a multi-layer atmosphere-soil-vegetation model with aerosol dynamics and dry deposition. In particular, the authors hope to explain observed apparent fluxes of NO3-, NH4+ and HNO3 above vegetative canopies by within-canopy evaporation of ammonium nitrate (NH4NO3).

In general, I believe this to be important work and interesting, valuable data, and I also agree with their major conclusion that 3-D chemical transport models need to better incorporate the within-canopy aerosol dynamic/equilibrium processes that are the focus of this work.

**Response: We sincerely appreciate your interests and positive comments on our manuscript.**

However, I believe that this article needs additional polishing to make its presentation more effective and the results more accessible to ACP readers. I offer my suggestions for this presentation enhancement below. I recommend that the article be published after additional measures are taken to enhance its presentation. *Response:* Thank you so much for helpful suggestions; we revised the manuscript as follows. We hope that the manuscript is drastically improved.

**Specific Comments:**

One concern is that the model used in this study is not adequately described. The paper does reference other published articles (Katata et al., 2013; Katata et al., 2014) where portions of the model are described in some detail; however, the full model used here seems to have been described in a gray literature document (Katata and Ota, 2017). It would be better if the authors included more model description in this paper, especially providing information on model setup for this particular application (e.g., model inputs, number of layers, time resolution, model outputs, etc.), referring to the other publications (or an Appendix) for details.

*Response:* Since a large part of description in Katata and Ota (2017) consists of our two published articles (Katata et al., 2013; 2014), the model has been basically reviewed in scientific journals. As suggested by the reviewer, we added the summary of simulation settings as supplemental table (Table S1) in addition to subsection 3.2.

A second major concern is the presentation of some of the Figures, as follows:  $\bigcirc$  Figures 2 & 3 – The figures are very small and cannot be adequately evaluated, especially with respect to

the agreement between the measurements and model results. Some way needs to be found to present the figures in a larger, clearer way.

*Response:* As you suggested, these figures were too small for evaluation. We increased axis fonts of old Figs. 2 and 3, and separate to two figures as new Figs 2-5.

 $\bigcirc$ Figure 5 – The colors chosen for the vertical profiles are very difficult to distinguish between some species. Bolder color differences would be a great improvement in understanding this very important figure. *Response:* The figure (new Fig. 7) was revised with different colors which enable readers to understand.

○Figure 9 – The y-axis title is so small as to be illegible. Please increase. *Response:* As you suggested, the font of both axes of new Fig. 11 was increased with a modification of alignment.

 $\bigcirc$ p. 4, lines 90-91, line 104, line 111: There seem to be two definitions of "a", one which I believe is the leaf area density, and the other a constant in Eq. (4).

*Response:* Those confused the reviewer. We defined "a" as the leaf area density in Eqs. (1) and (2), and new "b" as the constant for Eq. (4).

○p. 4, lines 90-91: The term "R" is not defined.
○p. 4, line 104: "Tc" is not defined.
Response: We defined the above variables.

 $\bigcirc$ p. 4, Eq. (4): A reference should be provided for this formula, which is Massad et al. (2010), ACP, 10, 10359-10386.

Response: We added the reference as suggested.

○p. 6, line 148: Was it really a "grass" fiber filter or was this supposed to be a "glass" fiber filter? *Response:* It was typo; we corrected as "glass" (L.171, p.6)

Op. 8, Section 3.3: The description of the simulation scenarios is somewhat confusing on first reading. The phrases "NH4NO3 equilibrium" or "no NH4NO3 equilibrium" might lead someone to believe that nonequilibrium thermodynamics is being modeled here, when actually it's just that no NH4NO3 gas-particle exchange is being allowed in the "no NH4NO3 equilibrium" scenario. A possible suggestion might be something like "NH4NO3 gas-particle conversion" and "no NH4NO3 gas-particle conversion". *Response:* Since our wording was confusing, all of NH4NO3 equilibrium was replaced to "NH4NO3 gas-particle conversion".

○p. 4, line 96: Should be "perfect absorption".

Op. 6, line 150: If I understand the intention of the sentence, both instances of "reading" could (and should)

be deleted – "We obtained 5 daytime data sets and 6 nighttime data sets."

○p. 8, line 235: Should read "... on both NH3 and fine NH4+ concentrations ..."

○p. 8, line 239: Should be "... competing shrinkage mechanism, ...".

○p. 9, line 252: Should be "… evaporation has less impact on

○p. 9, line 269: Should be "among", not "amoung".

Op. 11, line 334: There are two instances of "typically" in this sentence, which is awkward.

Op. 12, in "Author contributions": Should be "... developed the model with support from MK,".

*Response:* All items were revised. Thank you for your suggestions.

○p. 11, first paragraph in Section 5.3: This discussion here (and Figure 10) is very confusing. How can the ratios be plotted as a function of RH, but distinctions still made between "high RH conditions" and "low RH conditions"? Whatever the subject is here, it is needs to be more clearly explained.

Figure 10 - As mentioned below, this figure (and the discussion that goes with it) is very confusing. I don't really understand what point is being made with this figure (ratios as a function of RH, but under "high RH conditions" and "low RH conditions" – this doesn't make sense as explained).

*Response:* We should explain about the background of the figure in more detail. We added the sentence about "In this study, since water uptake of aerosols, typically represented as the hygroscopic growth factor defined as the ratio between the humidified and dry particle diameters, is almost negligible under RH < approximately 80 % and increases over RH > 80 % (e.g., Fig. 6 in Katata et al., 2014), we defined the threshold of 80 % for high and low RH conditions." (L.362-365, p.12).

**Title: Aerosol dynamics and gas-particle conversion in dry deposition of inorganic reactive nitrogen in a temperate forest Authors: G. Katata et al.**

**Author response to reviewer comments**

**Response to Anonymous Referee #2**

The paper presents a 1-dimensional model of coupled phase-partitioning and multi-layer canopy to study the process of NH4NO3 aerosol evaporation and water uptake on surface atmosphere exchange fluxes of reactive nitrogen compounds. A number of studies have observed remarkably high apparent deposition rates of ammonium and nitrate aerosol which have been linked to aerosol evaporation processes, but few studies have attempted to model the process and thereby quantify the overall effect on N deposition. This paper presents a detailed model for this purpose and applies to reproduce concentration gradients measured above a Japanese forest. Whilst the model is detailed, the measurements are comparably basic and poorly time-resolved which limits their value in really testing the model. Nevertheless, the paper presents a useful first application of the model and makes a valuable contribution to the literature on the subject. I recommend its publication once a few concerns have been addressed as detailed below.

Response: We sincerely appreciate your interests and positive comments on our manuscript. In the original manuscript, we should discuss about the model results via the uncertainties of measurements more carefully as a first application (not for model validation). Therefore, as responded below we revised the whole sentences throughout the manuscript regarding the input data based on your comments and suggestions. We hope the manuscript should be clearly improved. For example, we inserted the sentences about the uncertainty of the datasets: "In this study, we made a simulation with basic and less time-resolved datasets as very first application of the model to the NH4NO3 gas-particle conversion and aerosol water uptake of reactive nitrogen compounds. The above uncertainties associated with input data such as number concentration and particle size distribution should be improved in future." (L.239-241, p.8).

Importantly, the description of the model and its application is very brief and important detail is missing. In addition, the authors could exploit their results a little more comprehensively. My suspicion is that this would result in a sufficient amount of new, additional text to necessitate a re-review of the manuscript. *Response:* A part of model description and simulation conditions were not enough for evaluation. As responded to

your suggestions and comments below, the manuscript was drastically improved. Thank you to your comments.

**Major concern**

1. My major concern relates to the use of long-term average concentrations. Please make the time resolution of

the filter pack measurements more explicit. It suggests in Line 147 that integration times were 9 hrs (day-time) and 15 hrs (night-time), whilst line 284 talks about weekly measurements. In either case I do not understand why they were used as spot measurements in the interpolation (Figs. 1 and 4). Should the model not be initialised with the same concentration for those time-intervals? Otherwise the average of the interpolated time-series does not match the measured average.

Response: The description about filter-pack sampling was confusing and caused misunderstanding of reviewer. For the early autumn period, filter-pack sampling was continuously made during the day and night for no rain days. For the late autumn period, the time resolution was relatively low as weekly continuous measurements. However, the data in rain days of the early autumn period were missing and caused gaps as shown in old Figs. 1 and 4. As a result, we had no choice to make the data of these gap periods linearly incorporated for simulations. As you suggested, since this interpolation could cause unrealistic effects on our results and should not be analyzed, we shaded the output data during the rain period (no available filter-pack data) in new Fig. 6. In the original manuscript, this error should not affect our results (old Fig. 5) because we used only the calculations and measurements in the above no rain period for comparisons of inorganic mass concentration. Regarding initialization, 12 hour was applied to spin-up time in this study, which is considered to be sufficiently low the gas and particle exchanges between atmosphere and vegetation was sufficiently higher than time resolution of filter-pack data (~ half day). To address this, we revised the sentences as "For the early autumn period, filter-pack sampling was continuously performed during the day and night except when it was raining. As a result, 5 daytime reading data sets and 6 nighttime reading data sets were available. The gaps between data in rain days of the early autumn period were linearly incorporated for simulations. Since the interpolation could cause unrealistic effects on the results, we used only the calculations and measurements in the above no rain period for comparisons of inorganic mass concentration. For the late autumn period, the time resolution was relatively low as weekly continuous measurements." (L.172-177, p.6) and "The model profiles averaged for only the sampling periods were compared with observed ones." (L.276, p.9).

Similarly, the length of the intervals of the dots in Figure 4 do not suggest that the measurements were continuous. Line 251 and Figure 6. I would be very surprised if the conditions for NH4NO3 condensation were met in this forest. Normally, these conditions are found over strong sources of NH3 (e.g. Nemitz et al., 2009). If I understand the manuscript correctly, the authors are using long-term average concentration measurements with long-term average meteorology. In general, the vapour pressures of NH3 and HNO3 are dependent on temperature and humidity in a highly non-linear relationship. This means, if long-term average concentrations are paired with long-term averages in temperature and humidity, it is unlikely that the comparison of measured concentrations with the thermodynamic equilibrium concentrations evaluated with the ISORROPIA2 model can correctly assess whether there is potential for NH4NO3 evaporation of condensation. With the long-term integrated samples this problem cannot be fully resolved and as a result the entire manuscript needs to be reformulated to some degree that full agreement between modelled and measured conditions cannot be expected. The application to this measurement dataset can only be considered a first test of the model, rather

than a conclusive assessment of its capability. This problem needs to be discussed and, potentially, a sensitivity analysis could explore the uncertainty introduced by the averaging. In addition, the concentrations used for initialisation are subject to measurement uncertainty that may further limit the model / measurement comparison.

*Response:* Again, our description was unclear and caused misunderstanding of the reviewer. The input filter-pack data of was half-daily with linear interpolation (early autumn period) or weekly (late autumn period), but the output data of all mass and number concentrations was half-hourly as well as all meteorological variables (old Fig. 3). These were averaged for only the sampling periods for comparisons with observations. Therefore, the effect of diurnal changes of air temperature and humidity should be reflected to the results in the early autumn period, which is not so problematic, we believe. In order to emphasize that this work is a first application of the model, the details in uncertainties are added based on your suggestions: "Several uncertainties (e.g., low time resolution of weekly filter-pack data in the late autumn period; initialization of measurement uncertainty; complex topography of the study site) may cause underestimations in calculated wind speed (Fig. 3a and 4a) and overestimations in total number concentration within the canopy after 25 November 2016 (Fig. 9a). In Fig. 8, the conditions for NH4NO3 condensation were calculated in this forest, although these conditions are normally found over strong sources of NH3 (e.g. Nemitz et al., 2009). Thus, the results from this study can only be considered a first test of the model to the NH4NO3 gas-particle conversion and aerosol water uptake of reactive nitrogen compounds, rather than a conclusive assessment of its capability." In addition, the name of subsection was also revised as "Uncertainties in observation and model results". (L.324-331, p.11).

2. Overall, I am missing results and discussion on the effect of gpc (and also the equilibration with water content) on the exchange with the vegetation, in addition to the effect on the fluxes above the canopy. Only if this effect is shown to be significant, would there a need to incorporate this additional complexity into deposition schemes. To assess the importance, I would encourage the authors to quantify, from their results, the effect on the actual ecosystem flux of the various forms of N, total reactive N and also the effective bulk deposition velocity of the aerosol at the surface (Vds), which changes because particle size changes. Presumably, this change in Vds is the reason for gpc changing also the in-canopy gradient of SO42. (Fig. 5) although it does not take part in the gpc process itself. The implications should be discussed.

Response: Thank you so much for your crucial suggestions. We calculate the changes in apparent mass flux over the canopy and the contribution of each compounds on total nitrogen flux. We also showed the vertical profiles of mass flux as well as concentration (Fig. S2). To demonstrate the impact of gpc on particle (especially NO3-) deposition flux, the following discussion was inserted: "Once the gpc process is considered, particle deposition could have very important contribution as nitrogen flux over the forest ecosystem. Comparing the calculated daytime mass flux at 30 m height between "no gpc" and "gpc" scenarios in the early autumn period (Fig. S2),

deposition flux of fine NO3- and NH4+ was 15 and 4 times higher in "gpc" scenario. Since there was almost no change in SO42- flux between two scenarios, this change is only caused by gpc. For gas species, both HNO3 and NH3 slightly decreased to 0.6 and 0.8 times due to evaporation of NH4NO3 particles. This change in flux could be applied to that in deposition velocity of each species. Furthermore, although particle deposition flux contributes to only 5 % of total nitrogen flux above the canopy in "no gpc" scenario, this impact was increased to 38 % (NO3-: 27 %, NH4+: 11 %) in "gpc" scenario. It should be noted that the contribution of NH3 was still large as 37 % in total nitrogen flux even in the "gpc" scenario. The above results indicate that the increase of (apparent) particle deposition due to NH4NO3 evaporation may be important in the chemical transport modeling.". (L.391-400, p.13). Part of this explanation also appeared in Abstract.

3. Related to this, the discussion of Figure 9 is very cryptic and only accessible to those already very familiar with the subject. It is closely linked to the observations of apparent bi-directional size-segregated fluxes (e.g. Nemitz and Sutton, 2004; Ryder, 2010) and this link should be made.

Response: Thank you for introducing Ryder (2010). We knew the paper from Nemitz (2015), but we did not refer it because we could not have an access to the document as it is the doctoral thesis. As you suggested, we incorporated additional explanation as "This is a similar result to demonstrated by past numerical study for size-resolved particle number flux (Ryder, 2010); the certain particle diameter around 0.15 µm at which the apparent flux switched from deposition to emission within the canopy and approximately reflected the peak in the number size distribution. Furthermore, the apparent emission flux was represented as more particles shrink into a given size bin from the next larger size than are leaving the bin to the next smaller size, while more particles shrink out of a given size bin than shrink into it from the next larger size bin, resulting in apparent fast deposition (Ryder, 2010)." (L.309-314, p.10-11).

Additional scientific comments and needs for clarification:

 $\bigcirc$ Abstract. It would be helpful to be more quantitative and also to include a statement on the effect on the NH3 flux as well as the total reactive N flux. By what fraction does the change in phase partitioning change the net N flux?

*Response:* Thank you for your suggestion. We calcualted the contribution of all reactive nitrogen components (including NH3) to total nitrogen flux as "Once the gpc process is considered, particle deposition could have very important contribution as nitrogen flux over the forest ecosystem. Comparing the calculated daytime mass flux at 30 m height between "no gpc" and "gpc" scenarios in the early autumn period (Fig. S2), deposition flux of fine NO3- and NH4+ was 15 and 4 times higher in "gpc" scenario. Since there was almost no change in SO42- flux between two scenarios, this change is only caused by gpc. For gas species, both HNO3 and NH3 slightly decreased to 0.6 and 0.8 times due to evaporation of NH4NO3 particles. This change in flux could be applied to that in

deposition velocity of each species. Furthermore, although particle deposition flux contributes to only 5 % of total nitrogen flux above the canopy in "no gpc" scenario, this impact was increased to 38 % (NO3-: 27 %, NH4+: 11 %) in "gpc" scenario. It should be noted that the contribution of NH3 was still large as 37 % in total nitrogen flux even in the "gpc" scenario. The above results indicate that the increase of (apparent) particle deposition due to NH4NO3 evaporation may be important in the chemical transport modeling." (L. 391-400, p.13). Part of this explanation also appeared in Abstract.

OLine 65. Please clarify if the model also predicts the relative humidity profile which is important in controlling the phase partitioning. Related to this, Figure 3 should convey better whether the measured (in-canopy) profiles in the meteorological parameters are correctly reproduced by the model, i.e. the emphasis should be on the vertical change rather than the time-series.

*Response:* Since the relative humidity profile was predicted in the atmosphere sub-model, we added the word "relative humidity" as well as specific humidity at the line 63 in p.3. Furthermore, as you suggested in your main concern, we should emphasize that this was not model validation study but very first application to using limited datasets. Thus, this study should not demonstrate agreement between calculations and observations (as performance test), so that we should much focus on comparisons as first application of the model to the NH4NO3 gas-particle conversion. As a result, we added new Fig. S1 for vertical profiles of meteorological variables and revised the sentences as "For calculated air temperature and humidity, the primary determinants of ambient conditions of gas-particle conversion and aerosol hygroscopic growth, calculated temporal changes were closed to the observations within the canopy (Fig. 3 and 4; b and c). The above features were found in mean vertical profiles during the daytime and nighttime (Fig. S1)." (L.258-261, p.9).

OLines 100ff. I do not understand the approach taken for calculating  $\chi d$ . The text reads as if this is chosen to match the atmospheric concentration at each canopy layer. Surely, in this case Fgd becomes zero if the canopy layer air concentration matches the gas phase concentration in equilibrium with the  $\Gamma d$  of the leaf water layer, and rd ceases to have any effect. It is exactly the departure from equilibrium that drives the flux. Instead,  $\Gamma d$  is controlled by the previous accumulated deposition onto the leaf cuticle and the size of the water pool. How is the water pool size calculated in the model? Related to this, I am not convinced the use of an rd that is linked to acid/NH3 ratio and a leaf water emission potential are internally consistent. Parameterisations of the cuticular resistance as a function of this ratio have been developed and applied within the framework of a zero leaf water emission potential (e.g. Nemitz, 2015; Fowler et al., 2009) meant to account for the effect of a non-zero leaf water concentration. By additionally introducing a non-zero  $\Gamma d$  value, this effect is accounted for twice.

Response: The model description related to  $\chi d$  for wet canopy was not enough. We incorporate the sentences how to calculate leaf surface water from Katata et al. (2013) as follows: "The RH value could be affected by the leaf surface water content predicted at each canopy layer, based on water balance due to evaporation of leaf surface

water, interception of precipitation by leaves, capture of fog water by the leaves, and the drip from leaves (Katata et al., 2008; 2013)." (L.118-121, p.4-5). The surface water capacity of 0.2 kg m-2 was given as leaf surface water amount where there is maximum evaporation by Deardorff (1978). The NH3 exchange processes scheme has been validated at rice field in Japan (Katata et al., 2013), but as you suggested, there is the uncertainty for applying rd to the condition of non-zero leaf water concentration of NH3. Thus, we added the sentences as "Since our model is not the dynamic modeling approach (e.g., Sutton et al., 1998; Flechard et al., 1999) to simulate NH3 charging and discharging of the cuticle, Equation (4) could have uncertainty at wet canopy in equilibrium with non-zero leaf surface concentration of NH3." (L.121-123, p.5).

Deardorff, J.W., 1978. Efficient prediction of ground surface temperature and moisture, with inclusion of a layer of vegetation. J. Geophys. Res. 83, 1889–1903.

OLine 113. I realise that the authors are here only summarising the principles of a paper that is described in more detail elsewhere. Nevertheless, it would help to cite the approach taken to estimate the aerosol capture efficiency. Similarly, the origin of Eq. (4) needs to be mentioned.

*Response:* We added the original references for Eqs. (4) (Massad et al., 2010) and (6) (Fuchs, 1964; Kirsch and Fuchs, 1968; Peters and Eiden, 1992; Petroff et al., 2009).

OLine 130. In addition to the ISORROPIA2 thermodynamic module, does the model treat any gas-phase chemistry? This may be important as an additional source for HNO3. If not, the authors should discuss the implications somewhere.

*Response:* The gas-phase chemistry such as HNO3 production from NO2, HONO, N2O5, and so on was not considered in the model. Since the process could cause uncertainties in our results as you suggested, we added the sentence as "The gas-phase chemical production of HNO3 could affect the simulated HNO3 concentration and flux, which should be implemented to the model in future." (L.153-154, p.6).

OLine 198. It is not clear how the leaf water content was prescribed. This should affect the overall RH profile throughout the canopy and thus the results should be quite sensitive to this parameter?

*Response:* Since the description about leaf water content calculation was missing in the manuscript, we added the sentences "The RH value could be affected by the leaf surface water content predicted at each canopy layer, water balance due to evaporation of leaf surface water, interception of precipitation by leaves, capture of fog water by the leaves, and the drip from leaves. (Katata et al., 2008; 2013)." (L. 118-121, p.4-5). This could increase the RH value in the canopy layer, although test periods for comparisons between observations and calculations were without rainfall (L.175-176, p.6).

OLine 203. The meaning of fio is not clearly introduced. I understand it to be the ratio of inorganic to total aerosol mass. If so, it is constrained by the observations (in contradiction to what is stated in the manuscript) by the comparison of total inorganic aerosol mass (from the filter-pack measurements) to total aerosol mass (approximated via the ELPI+ aerosol volume). What is not constraint is its size dependence.

*Response:* The description about fio estimation was confusing and further information about ELPI+ data was required. Your understanding about calculating fio for fine and Aitken modes is correct, but we required to assume its values for both fine and Aitken modes because the ELPI+ data (total mass concentration) was limited for only the late autumn period. Since the sentences were not organized, we revised them as follows: "In order to simulate the vertical profiles of total number concentration within the canopy, the volume fraction of inorganic compounds, fio, was given by the data of total inorganic and total mass concentrations. For the late autumn period, temporal changes in weekly fio values for fine and Aitken modes was given by the data of filter-pack and ELPI+ measurements. However, for the early autumn period, the ELPI+ measurements were unavailable as described in last subsection. Therefore, temporal changes in fio (Fig. 1b) were set based on the filter-pack data at the study site and total PM2.5 mass concentrations observed at the nearest air quality monitoring station at Hachiouji (3 km west-north-west from the site). For both periods, fio for Aitken mode was assumed to be the same as that for fine mode, since again no observational data were available." (L.229-236, p.8).

OSection 3.2. I am missing more explanation as to how the measurements were used to drive the model and provide initial or boundary conditions. In fact the content of this section does not match its title. Did you use concentrations at a single height (if so, which one?) or several heights? How often was the model re-initialised with the measurements. Was it allowed to run to steady-state conditions or was it continuously perturbed by the measurements? Was a spin-up time used? Maybe, the input could be illustrated by adding the constraining concentrations as a top panel to Figure 4.

Response: We should show detailed explanation about initial and boundary conditions. To present it more clearly, new Table 1 was inserted for both periods. No spin-up time was applied in this study because the gas and particle exchanges between atmosphere and vegetation was sufficiently higher than time resolution of filter-pack data (~ half day).

○In addition, it is numerically problematic to use the same aerosol composition across all sizes as the Kelvin effect then causes evaporation from the smallest particles and condensation on the larger ones. This would drive some of the changes in the diameters (Fig. 7), which would then not represent a response to vertical gradients but reflect inadequate initial conditions. How was this problem dealt with? In this context (and in general) I would encourage the authors to study and refer the work of Ryder (2010) who also developed a similar model and applied it to existing datasets of exchange.

Response: Since we are using a modal aerosol dynamics method, differences in Kelvin effect for fine and Aitken

mode particles are considered by using their mass mean diameter to represent the effect, but those within a mode cannot be resolved. This should be problematic as the standard deviation of the size distribution is enhanced. We somehow dealt with it by applying mode merging (or mode renaming), a part of Aitken mode (or accumulation mode) shifts to accumulation mode (or Aitken mode) when condensation (swelling) occurred to avoid unrealistically broad size distribution (e.g. Kajino et al., 2012). Certainly, our assumption, same composition in size at the initial and boundary conditions might cause large uncertainty in simulated vertical size distribution profile and its perturbation, but the size-resolved measurements have not been available so far. In future, we need to conduct size-resolved composition measurements to assess this uncertainty by using size-resolved numerical model (i.e., bin model).

Thank you for introducing Ryder (2010). We knew the paper from Nemitz (2015), but we did not refer it because we could not have an access to the document as it is the doctoral thesis. Consequently, in the revised manuscript, we added the following statements in subsection 5.1 with referring Ryder (2010): "Another uncertainty in the results could be associated with same composition assumption in size at the initial and boundary conditions. Variations of chemical composition in size caused variations in equilibrium vapor pressure at particle surface due to Kelvin and Raoult's effects, which caused uncertainty in the simulation of swelling and shrinking of particles. Since we used a modal aerosol dynamics method, differences of these effects within each mode cannot be resolved. We need to revisit this issue in the future by using size-resolved composition measurements and size-resolved aerosol model as done by Ryder (2010) to assess this uncertainty." (L.332-337, p.11).

 $\bigcirc$ Line 210. Values of cg and cs as high as 300 ppb and 2000 ppb are completely unrealistic and inconsistent with the paper of Massad et al. (2010). They also ignore the temperature dependence of Eq. (3). Do the authors mean Gg = 300 and Gs = 2000 (no units!)? Gg is mentioned for the first time here and needs to be introduced much earlier on. Again, this calls for introducing the model in more detail than is currently done.

*Response:* The variables  $\chi g$  and  $\chi s$  were wrong and confusing the reviewer. These should be  $\Gamma g$  and  $\Gamma s$  (without unit), respectively. We modified the above variables and introduced them at new Eqs. (5) and (6), and modified.

OSection 5.1 and Line 349ff. Figure 4 suggests to me that the extrapolation of concentration into the canopy led to better agreement in the run that did NOT include gpc, whilst the text talks about an improvement. Please clarify. In general, to assess model performance, it would be much more illustrative to rearrange Figure 5 so that the three lines (Obs, no gpc, gpc) can be compared on a single plot for each compound. In addition, it would be interesting to compare a plot of how fluxes changes with height.

*Response:* As you suggested in your main concern, we should emphasize that this was not model validation study but very first application to using limited datasets. Thus, this study should not demonstrate model improvement and performance test, so that we should much focus on comparisons as first application of the model to the NH4NO3 gas-particle conversion. The sentences in Conclusions were now revised as "In the simulation including NH4NO3 gas-particle conversion processes, vertical gradients of normalized mass concentrations of nitrogen gases (HNO3)

and NH3) and fine particles (NO3- and NH4+) within the canopy were clearly high than that for SO42-." (L.418-420, p.14). In section 5.2, the profiles of daytime flux were introduced as new Fig. S2 for all inorganic nitrogen components as "The above feature was also apparent in vertical profiles of mass flux for all inorganic nitrogen components during the daytime (Fig. S2)." (L.260-261, p. 9).

OLine 326. I am not convinced the authors' argument here is correct. I would expect, to the first order, HNO3 and NH3 to be driven off the aerosol in stoichiometric ratios and thus the flux divergence for both compounds (and their aerosol counterparts) should be similar in absolute (molar) terms, independent of the deposition rate of the individual compounds. However, Fig. 5 shows normalised (i.e. relative) concentration changes and here the authors are correct with their second explanation: the relative effect on NH4+ is smaller than on NO3-because it partly represents non-volatile sulfates.

*Response:* Our statements were somewhat confusing. We removed it and added the sentences accordingly: "NH4+ flux over the forest was less influenced than NO3- by gas-to-particle conversion. It is because dry deposition rates of NH3 were much lower than those of HNO3, and so differences in deposition rates between NH3 and NH4+ are much smaller than those between HNO3 and NO3." (L.382-384, p.13).

**Technical corrections:**

○Title. The word "in" does not read right in my mind. How about "The effect of aerosol dynamics and gas-particle conversion on dry deposition …"

*Response:* Thank you for your suggestion: I modified the title as "The effect of aerosol dynamics and gas-particle conversion on dry deposition of inorganic reactive nitrogen in a temperate forest".

OLine 88. "fluxes with stomata ... and with leaf water surfaces ..." *Response:* We replaced "over" to "with" as you suggested.

 $\bigcirc$ Eqs. (1), (2), (3), (5). Please make sure the text introduces all symbols used in the equations. Many symbols (e.g. R, a, Gs) are not introduced.

*Response:* The variables are not introduced as you suggested. We checked the equations and added the explanation for a, R', Gs (with the appropriate reference; Nemitz et al., 2004).

**$\bigcirc$ Eq. (3) would benefit from a reference.**

Response: We gave two references for this equation (Nemitz et al., 2000; Sutton et al., 1994).

 $\bigcirc$ Eq. (5). Is the meaning of a here the same as in Eqs. (1) and (2)? If not, please use a different symbol. *Response:* The explanation of "a (leaf area density)" was missing; as other reviewer also suggested, we introduced this variable just after Eqs. (1) and (2).  $\bigcirc$ Eq. (5) and (6) and associated text. It would help here to make any dependence on particle diameter explicit, e.g. by writing Ep(Dp).

*Response:* As you suggested, we modify both equations (currently Eqs. (7) and (8)) with dependence of particle diameter (Dp) with an explanation.

OLine 130. "... transfer is driven by the difference ..." Response: We modified this sentence as you suggested.

OLine 173. It is not clear to me what the word "latter" refers to. Please rephrase. *Response:* The word "latter" was rephrased as "incoming long-wave radiation". (L.199, P.7)

OLine 190. It would be useful to state the total number of layers of the model. Response: As suggested by other reviewer, we added the information of layer number in Table S1 in supplement.

OLine 214. The acronym "gpc" is not introduced. Presumably it stands for gas-particle conversion? I wonder whether "thermodynamic gas-particle partitioning" would be a better concept to use throughout? *Response:* We should introduce the acronym in the manuscript. As you suggested, we define the word gpc as "thermodynamic gas-particle partitioning" (L.XX, p.YY).

 $\bigcirc$ Figure 6. Please clarify in the figure caption the height for which this flux is provided as the flux is height dependent. In addition, it may be illustrative to display the actual exchange with the ecosystem under both model scenarios.

*Response:* We added the height (30 m) for output of (apparent) mass flux in the caption of new Fig. 8. Furthermore, "The actual deposition flux of each component by forest (ecosystem flux) was also shown for comparisons with apparent flux." (L.42, p.2).

OAll figures: horizontal zero lines on all figures would help interpret these more easily. The font size of some labels and legends should be increased for better readability.

*Response:* We increased font sizes and added zero lines to all figures.

OLine 322. This should read "2004" rather than "2004a" here, I believe. *Response:* It was corrected to "2004". Thank you so much.

**Title: Aerosol dynamics and gas-particle conversion in dry deposition of inorganic reactive nitrogen in a temperate forest Authors: G. Katata et al.**

**Author response to reviewer comments**

**Response to Anonymous Referee #3**

General The author present the development of an advanced deposition model, here for the inorganic reactive nitrogen gas phase species HNO3 and NH3 and particle species NO3- and NH4+. I suggest to use not use the term 'aerosol' when solely particles are addressed. Use the term 'aerosol' when you address particle together with the gas phase where they are dispersed in. Otherwise use 'particle'.

*Response:* Thank you for your suggestion. As you suggested, we replaced most of words of "aerosol" into "particle" throughout the manuscript.

Surely, deposition much deserves a better treatment in many atmospheric model, so in principle an improvement in deposition schemes is highly welcome.

**Response:** We appreciate for your positive comments on our work.**

Overall, I feel the model can deliver useful results but there are many approximations it its set-up. This should be treated most carefully. In my view, the paper needs a huge amount of improvement but I rate this as still doable and not recommend rejection. I would therefore like to recommend major revision according to all reviewer comments with external re-review necessary.

*Response:* Thank you for your interests and many suggestions on the manuscript. As you suggested, we should have emphasized that this was fisrt application by clearly stating the model description and setup for numerical experimental.

**Details**

OAbstract: I feel the abstract should give more information, at best, in a quantitative manner. It now reads too much like an introduction. What is the main numerical outcome? What is better than before? The abstract should clearly state what is treated.

*Response:* As you suggested, the information was not enough for readers. We totally revised the sentences and added more sentences in Abstract as follows: "Although dry deposition has an impact on nitrogen status in the forest environments, the mechanism for high dry deposition rates of fine nitrate particles (NO3-) observed in forests remains unknown and is a potential source of error in chemical transport models. Here we modified a multi-layer land surface model coupled with dry deposition and aerosol dynamics processes for a temperate mixed forest in Japan, so that we carried out its first application to the ammonium nitrate (NH4NO3) gas-particle conversion (gpc)

and aerosol water uptake of reactive nitrogen compounds. The processes of thermodynamics, kinetics, and dry deposition for mixed inorganic particles are modeled by a triple-moment modal method. The data of inorganic mass and size-resolved total number concentrations measured by filter-pack and electrical low pressure impactor in autumn was used for model input and numerical analysis. The model overall reproduces observed turbulent fluxes above the canopy and vertical micrometeorological profiles as our previous studies. The sensitivity tests with and without gpc demonstrated inorganic mass and size-resolved total number concentrations clearly changed within the canopy. The results also revealed that the within-canopy evaporation of NH4NO3 under dry conditions significantly enhances deposition flux for fine NO3- and NH4+ particles, while reducing deposition flux for nitric acid gas (HNO3). As a result of evaporation of particulate NH4NO3, the calculated daytime mass flux of fine NO3- over the canopy were 15 times higher in gpc" scenario than "no gpc" scenario. This increase caused high contribution of particle deposition flux to total nitrogen flux over the forest ecosystem (~ 38 %), while the contribution of NH3 was still large. A dry deposition scheme coupled with aerosol dynamics may be required to improve the predictive accuracy of chemical transport models for the surface concentration of inorganic reactive nitrogen.".

**OLine 13: Maybe a word should be added after 'nitrogen'? Like 'input'? *Response:* The word of "input" was inserted into the sentence (L.19, p.1)**

**OLine 15, 16: Why is these only Japanese references, please check other deposition work.**

*Response:* We should describe the background more in detail. To our knowledge, the references for "direct measurement of nitrate dry deposition" are rare except for Japanese work (c.f., Nakahara et al., 2019). To emphasize this, we revised the sentences with several references of dry deposition estimate studies in East Asia as "In East Asia, where air pollutant emissions continue to increase (EANET, 2016), although the importance of dry deposition of inorganic reactive nitrogen is suggested by prior studies by indirect estimate studies (e.g., Pan et al., 2012; Li et al. 2013; Xu et al., 2015), direct measurement studies are still limited (Nakahara et al., 2019). Recent observational studies at forests revealed that dry deposition flux of inorganic reactive nitrogen of fine NO3- was markedly higher than that expected from theory (Takahashi and Wakamatsu, 2004; Yamazaki et al., 2015; Honjo et al., 2016; Sakamoto et al., 2018; Nakahara et al., 2019)." (L.19-24, p.1-2).

**OLine 29: This is not only known from/for deposition studies but also for myriad of particle characterization studies. Are there more recent references?**

*Response:* We referred the textbook for well-known NH4NO3 volatilization process in atmosphere as "although the process itself has already been known in the atmospheric chemistry community for a long time (Seinfeld and Pandis, 2006)" (L.39, p.2). Regarding to the recent papers for deposition modeling, to our best knowledge only two papers are available as follows: Nemitz and Sutton (2004) (appearing from L.41 in p.2) and Ryder (2010) suggested by other reviewer (L.309, p.10).

Ryder, J., 2010. Emission, deposition and chemical conversion of atmospheric trace substances in and above vegetation canopies. PhD Thesis, University of Manchester, UK. Available from the University or via https://nora.nerc.ac.uk

**OLine 51: What does SOLVEG mean?**

*Response:* Our model "SOLVEG" was defined as "a multi-layer atmosphere-SOiL-VEGetation model" in the last manuscript (currently L.54, p.2)

**OLine 74: Reference Genuchten's concept**

*Response:* We referred the paper for unsaturated snow modeling as "Hirashima et al., 2010; Katata et al., recently accepted) (L.83, p.3).

**OLine 99: How does Eqn (3) relate to the Henry Constant? Can you clarify more what if written in the text?**

*Response:* Eq. (3) was the equation for calculating  $\chi$ s (stomata) and was not related to Henry's law. Since this paragraph was not organized and confused the reviewer, we insert the word "Meanwhile," before the sentence for  $\chi$ d which follows Henry's law and dissociation equilibrium with the atmospheric concentration of NH3 at each canopy layer." (L.111-112, p.4).

**OLine 104ff: Where does Eqn (4) come from? 'Affinity' is a strange term. Better justify the approximation for SO2 deposition.**

*Response:* Following the suggestion from other reviewer, we revised the sentence as "the following empirical formula for rd is applied" with the reference (Massad et al., 2010) (L.113, p.4). Although "affinity" was used in the original reference van Hove et al. (1989), we added the word as "(such as solubility on water)" to make its meaning clear (L.117, p.4).

**OLine 117 ff: See initial remark on nomenclature and revise this whole treatment consistent with clear naming.**

OLine 296: This headline must be revised. The size distribution does not have a formation mechanism, only the particles have

*Response:* The headlines were inappropriate; the word of "aerosol(s)" were revised to "particle(s)" when we reffered solely particles, based on your suggestion.

OLine 139, end: ... of the Tokyo ....
OLine 231: ... fine particles. *Response:* We corrected the sentences as you suggested.

OLine 148: What is a 'grass fiber filter'?

Response: This was a typo; the sentence was now revised as "glass fiber filter".

**OLine 200ff: There seem to be a lot of approximations for the particle size distribution initialization. How critical can this be for the overall study?**

Response: This sentence was inappropriate and confusing the reviewer. In fact, we made manual fitting independently using our ELPI+ data, which was compared with literature values as one of other examples at urban environment in autumn (Salma et al., 2011). To avoid this confusion, we revised this sentence as "The lognormal parameter sets of (Dg3,  $\sigma$ g) for fine and Aitken modes at the upper boundary condition were respectively set at (0.089 µm, 2.1) and (0.26 µm, 2.0) based on manual fitting of ELPI+ measurements at 30 m height (Fig. 1a). These parameter sets were applied to both the early and late autumn periods." (L.226-229, p.8). This uncertainty somewhat influences inorganic mass concentration profiles, but sensitivity tests with changing mean diameter showed that our important findings such as "vertical gradients of NO3- and NH4+ concentrations drastically increased due to NH4NO3 evaporation" (L.278-279, p.10) were consistent.

**OLine 361: I think feasibility might be the wrong term.**

Response: We deleted a part of the sentence "with high feasibility" since it was inappropriate.

**OFigures: It would be great to show correlation plots for some key properties rather than only time-series plots.**

*Response:* On the basis of the other reviewer, this study should not demonstrate model improvement and performance test based on scatter plots due to very limited datasets, so that we should much focus on comparisons as first application of the model to the NH4NO3 gas-particle conversion. This is emphasized in "In this study, we made a simulation with basic and less time-resolved datasets as very first application of the model to the NH4NO3 gas-particle conversion. The above uncertainties associated with input data such as number concentration and particle size distribution should be improved in future." (L.239-241, p.8)

**Aerosol The effect of aerosol dynamics and gas-particle conversion in on dry deposition of inorganic reactive nitrogen in a temperate forest**

Genki Katata1, Kazuhide Matsuda2, Atsuyuki Sorimachi3, Mizuo Kajino4, and Kentaro Takagi5

1Institute for Global Change Adaptation Science (ICAS), Ibaraki University, Ibaraki, 310-8512, Japan

2Tokyo University of Agriculture and Technology, 3-5-8 Saiwai-cho, Fuchu, Tokyo 183-8509, Japan

3Department of Radiation Physics and Chemistry, Fukushima Medical University, 1 Hikarigaoka, Fukushima, Fukushima 960-1295, Japan

4Meteorological Research Institute, Japan Meteorological Agency, Tsukuba, Ibaraki 305-0052, Japan

5Teshio experimental forest, Field Science Center for Northern Biosphere. Hokkaido University, Toikanbetsu, Horonobe, Hokkaido 098-2943, Japan

Correspondence: Genki Katata (genki.katata.mirai@vc.ibaraki.ac.jp)

Abstract. Although dry deposition has an impact on nitrogen status in the forest environments, the mechanism for high dry deposition rates of fine nitrate aerosols particles  $(NO_3^-)$  observed in forests remains unknown and is a potential source of error in chemical transport models. Here we developed a new modified a multi-layer land surface model coupled with dry deposition and aerosol dynamics processes for a temperate mixed forest in Japan, so that we carried out its first application

- 5 to the ammonium nitrate (NH4NO3) gas-particle conversion (gpc) and aerosol water uptake of reactive nitrogen compounds. The processes of thermodynamics, kinetics, and dry deposition for mixed inorganic aerosols-particles are modeled by a triplemoment modal method. The new model data of inorganic mass and size-resolved total number concentrations measured by filter-pack and electrical low pressure impactor in autumn was used for model input and numerical analysis. The model overall reproduces observed turbulent fluxes above the canopy and vertical micrometeorological profiles , as well as as our previous
- 10 studies. The sensitivity tests with and without gpc demonstrated inorganic mass and size-resolved total number concentrations clearly changed within the canopy. Sensitivity tests The results also revealed that the within-canopy evaporation of ammonium nitrate (NH4NO3) under dry conditions significantly enhances deposition flux for fine NO3- and NH4+ aerosolsparticles, while reducing deposition flux for nitric acid gas (HNO3). As a result of evaporation of particulate NH4NO3, the calculated daytime mass flux of fine NO3- 
[revised manuscript text omitted]

(2)

where a is the leaf area density (m2 m-3), Dgas and Dw are the diffusivities (m2 s-1) of trace gas and water vapor, rb, rs, and rd are the resistances (s m-1) for leaf boundary layer, stomata, and the evaporation (cuticular), χa is the ambient gas concentration (nmol m-3) in the canopy layer, and R' = (rbrs + rbrd + rsrd). The total gas exchange flux over the leaves can
be calculated as the sum of Fgs and Fgd for all canopy layers. In accordance with a number of observations (e.g., Huebert and Robert, 1985), all χs, rd, and rs are set to be zero for highly reactive and water-soluble gas species of HNO3 and HCl, i.e., perfect absorption by plant canopies. For both species, the parameterization for a deciduous forest by Meyers et al. (1989) is used to calculate rb. For NH3, χs is calculated based on the thermodynamic equilibrium between NH3 in the liquid and gas phases ÷

110 (Nemitz et al., 2000; Sutton et al., 1994):

$$\chi_s = \frac{161500}{T_c} \exp\left(\frac{10378}{T_c}\right) \Gamma_s,\tag{3}$$

where  $T_c$  is the canopy temperature (°C),  $\chi_s \Gamma_s$  is the stomatal emission potential (also known as the apoplastic ratio) at 1013 hPa . The (Nemitz et al., 2004). Meanwhile, the NH3 concentration at the leaf surface water ( $\chi_d$ ) is calculated by assuming Henrys Law and dissociation equilibrium with the atmospheric concentration of  $NH_3$  at each canopy layer. To calculate the exchange flux of  $SO_2$  and  $NH_3$  over the wet canopy, the following empirical formula for  $r_d$  is applied (Massad et al., 2010):

$$r_d = 31.5AR^{-1} \exp[b(100 - RH)],\tag{4}$$

where b is the constant, RH is the relative humidity (%) and AR is the ratio of total acid/NH3, represented as (2[SO2] + [HNO3] + [HCl])/[NH3] at each atmospheric layer. The value of AR is determined from calculations of gaseous inorganic concentration at each atmospheric layer. Since the affinity (such as solubility on water) of SO2 for the leaf surface is approx-

120

imately twice that of  $NH_3$  (van Hove et al., 1989), a half value of  $r_d$  calculated by Eq. (4) is applied to  $SO_2$  deposition. The *RH* value could be affected by the leaf surface water content predicted at each canopy layer, based on water balance due to evaporation of leaf surface water, interception of precipitation by leaves, capture of fog water by the leaves, and the drip from leaves (Katata et al., 2008; 2013). Since our model is not the dynamic modeling approach (e.g., Sutton et al., 1998; Flechard et al., 1999) to simulate  $NH_3$  charging and discharging of the cuticle, Eq. (4) could have uncertainty at wet canopy in equilibrium

**125 with non-zero leaf surface concentration of $NH_3$ .**

The exchange flux of NH3 over the ground  $(F_{q0})$  was described with compensation points at the ground  $(chi_q)$  as

$$F_{g0} = (D_{gas}/D_w)c_{H0}(\chi_{a0} - \chi_g),$$
(5)

$$\chi_g = \frac{161500}{T_{s0}} \exp\left(\frac{10378}{T_{s0}}\right) \Gamma_g,$$
(6)

where  $c_{H0}$  is the surface exchange coefficient for heat,  $\chi_{a0}$  is the NH3 concentration at the bottom of the atmospheric layer, 130  $T_{s0}$  is the soil surface temperature, and  $\Gamma_a$  is the emission potential at the ground surface.

As explained in Katata et al. (2014), the aerosol-particle deposition rate  $F_p$  ( $\mu$ g m-2 s-1 or # m-2 s-1) of each inorganic species in each canopy layer is represented as

$$F_p = aE_p(D_p),\tag{7}$$

$$E_p = \varepsilon(\underline{D}_p) F_f |\mathbf{u}| c_p(\underline{D}_p), \tag{8}$$

where Ep is the capture of aerosols particles by leaves (μg m-3 s-1 or # m-3 s-1); ε the total aerosol particle capture efficiency of plant leaves for aerosols particles by inertial impaction, (Peters and Eiden, 1992), gravitational settling, Brownian diffusion, and interception; (Kirsch and Fuchs, 1968), and interception (Fuchs, 1964; Petroff et al., 2009); Ff is the shielding coefficient for aerosols particles in the horizontal direction; |u| the horizontal wind speed (m s-1) at each canopy layer; and cp is the mass or number concentration of aerosols particles (μg m-3 or # m-3). Ep, ε, and cp are integration values of given size
bins with particle diameter (Dp [μm]).

**2.3 Aerosol dynamics**

In order to simulate changes in aerosol particle particle particle sizes due to condensation, evaporation, and water uptake, a triple-moment modal method (Kajino et al., 2012) is employed at each atmospheric layer in SOLVEG. Aerosols Particles are grouped into fine (accumulation) and Aitken mode with the size distribution prescribed by a lognormal function; while the

- 145 coarse mode is not considered in the simulation. The lognormal function is identified by three parameters: number concentration (N [# m-3]), geometric mean diameter ( $D_g$  [µm]), and geometric standard deviation ( $\sigma_g$ ). The triple-moment method predicts spatiotemporal changes in three moments (k) to identify the changes in the shape of each mode's lognormal size distribution. The selected three moments are 0th, 2nd, and 3rd moments ( $M_0$ ,  $M_2$ , and  $M_3$ ), which are respectively number (N), surface area (m2 m-3), and volume concentrations (m3 m-3).  $D_g$  values for each moment are named  $D_{g0}$ ,  $D_{g2}$ , and  $D_{g3}$ . The
- 150 relationship of the above lognormal parameters and the three moments for each atmospheric layer are as follows:

$$M_{k} = ND_{g0}^{k} \exp\left[\frac{k^{2}}{2}\ln^{2}\sigma_{g}\right],$$
(9)

$$D_{g0} = \left\lfloor \frac{M_2}{M_0} \right\rfloor^2 \left\lfloor \frac{M_3}{M_0} \right\rfloor^{-2}, \ln^2 \sigma_g = -\ln \left\lfloor \frac{M_2}{M_0} \left( \frac{M_3}{M_0} \right)^{-3} \right\rfloor.$$
(10)

Acrosol-Particle growth is dynamically solved in the same manner of Kajino et al. (2012). The gas to acrosol-particle mass transfer is accelerated driven by the difference between the current state and the thermodynamic equilibrium state, as simulated by ISORROPIA2 model (Fountoukis and Nenes, 2007) for semi-volatile inorganic components such as  $NO_3^-$ ,  $NH_4^+$ ,  $CI^-$ , and liquid water (H2O). The gas-phase chemical production of HNO3 could affect the simulated HNO3 concentration and flux, which should be implemented 
[revised manuscript text omitted]
 HNO3 and NO3. In fact, the observed deposition trends for NH3 and NH4+ were much weaker than those for HNO3 and NO3-. Also, while the major counter-ion of  $NO_3^-$  was  $NH_4^+$ , that of  $NH_4^+$  was not  $NO_3^-$  but  $SO_4^{2-}$ . Even though same count of molecules of  $NH_3$  and  $HNO_3$  evaporated, the gross deposition rate of  $NH_4^+$  appears to have been influenced mainly by  $(NH_4)_2SO_4$  and/or
- 400

 $NH_4HSO_4$  as also suggested by Nemitz (2015). The effect of  $NH_4NO_3$  equilibrium gas-particle conversion on  $NH_3$  flux was even lower than on fine  $NH_4^+$  (Fig. 4a6a) because the mass concentration of  $NH_3$  was much higher.

**5.4 Influencing the chemical transport modeling**

Once the gpc process is considered, particle deposition could have very important contribution as nitrogen flux over the forest ecosystem. Comparing the calculated daytime mass flux at 30 m height between "no gpc" and "gpc" scenarios in the early autumn period (Fig. S2), deposition flux of fine  $NO_3^-$  and  $NH_4^+$  was 15 and 4 times higher in "gpc" scenario, respectively. Since there was almost no change in  $SO_4^{2-}$  flux between two scenarios, this change is only caused by gpc. For gas species, both HNO3 and NH3 slightly decreased to 0.6 and 0.8 times due to evaporation of  $NH_4NO_3$  particles. This change in flux could be applied to that in deposition velocity of each species. Furthermore, although particle deposition flux contributes to only 5 %

410 of total nitrogen flux above the canopy in "no gpc" scenario, this impact was increased to 38 % (NO3-: 27 %, NH4+: 11 %) in "gpc" scenario. It should be noted that the contribution of NH3 was still large as 37 % in total nitrogen flux even in the "gpc" scenario. The above results indicate that the increase of (apparent) particle deposition due to NH4NO3 evaporation may be important in the chemical transport modeling.

Theoretical values of deposition velocity for sub-micron aerosols particles typically ranging from 0.1-1-1 cm s-1 may

- 415 have no substantial impact on surface concentrations in chemical transport models. However, as discussed in the previous subsection, high deposition velocity of fine  $NO_3^-$  due to evaporation in the forest (up to 40 times the above values) may effectively remove nitrate aerosols particles from the atmosphere over the forest and leeward. If the aerosol dynamics and gas-particle conversion processes can be incorporated into the dry deposition scheme in chemical transport models, we could improve upon or even eliminate prior studies' overestimates of the surface concentration of fine  $NO_3^-$  (Kajino et al., 2013;
- Shimadera et al., 2014, 2018; Morino et al., 2015; Sakurai et al., 2015). Hicks et al. (2016) found that in modeling deposition velocities of aerosolsparticles, the greatest uncertainty manifests in the range 0.1–1.0–1.0 μm. The cause of this uncertainty is still not convincingly established, although the differing treatments of some key aerosol-particle deposition processes (e.g., turbulent diffusion) have been suggested by prior studies (Petroff and Zhang, 2010; Zhang and Shao, 2014). As demonstrated in Fig. 10b–12b and c, evaporation of NH4NO3 under less humid conditions may play an important role for dry deposition of sub-micron aerosolsparticles.

**6 Conclusions**

A new multi-layer land surface model fully coupled with dry deposition and aerosol dynamics was developed to evaluate the impact of  $NH_4NO_3$ - $NH_3$ - $HNO_3$  conversion in temperate forests. The model was applied to field studies of mass and number concentration profiles in a Japanese mixed forest during autumn 2016. Four model scenarios with/without  $NH_4NO_3$  equilibrium

430 gas-particle conversion and/or aerosol water uptake were tested to quantify the impact of the above processes on dry deposition processes. While the The model overall successfully reproduced micrometeorological conditions (in particular, relative humidity) within and above the canopy, measured profiles of . In the simulation including NH4NO3 gas-particle conversion

processes, vertical gradients of normalized mass concentrations of nitrogen gases (HNO3 and NH3) and fine <del>aerosols</del> particles (NO3- and NH4+) within the canopy were <del>reproduced only in the model including NH4NO3 equilibrium processes. For aerosol</del>

- 435 clearly high than that for  $SO_4^{2-}$ . For particle size distribution, the observed emission tendency of total number concentration from the canopy to the atmosphere was explained by a larger effect of within-canopy evaporation of NH4NO3 than hygroscopic growth. As a result, the removal flux of calculated fine NO3- from the air above the forest to the forest can increase by up to 40 times under the *DRH* of pure NH4NO3. Similarly, the removal flux of calculated fine NH4+ can increase up to ~ 10 times, though calculations for fine NH4+ fluctuate strongly with *RH*. Conversely, HNO3 
[revised manuscript text omitted]

---

## Author Response (AR2)

**Title: The effect of aerosol dynamics and gas-particle conversion on dry deposition of inorganic reactive nitrogen in a temperate forest**
**Authors: G. Katata et al.**

**Author response to editor's comments**

Dear authors

  Please polish the English writing for this paper with help of an English native speaker or profession service before it can be accepted for publication.

*Response:* Dear Leiming Zhang, thank you so much for comment. We attached the new version with English correction. We hope this is accepted for publication at ACP.